# Joint-Space Empowerment as a Theory of Dexterous Motor Coordination

**James Heald** [1]  **Vittorio Caggiano** [2 3 4]  **Vikash Kumar** [2]  **Maneesh Sahani** [1]

## Abstract

Searching for effective policies is notoriously challenging in high-dimensional musculoskeletal systems, where multiple muscles actuate individual joints. Although this redundancy complicates naive policy search, it also implies that effective control can be captured by a low-dimensional action manifold. To identify such a manifold, we introduce Joint-Space Empowerment (JoSE), a novel information-theoretic objective that quantifies how much control an agent has over its mechanical degrees-of-freedom. We frame manifold discovery as an optimal precoding problem—where a state-dependent precoder maps low-dimensional latent actions to high-dimensional muscle commands—and derive the optimal precoder in closed form under control-affine Gaussian dynamics. We show that manipulation policies learned on this manifold display significantly enhanced dexterity, sample efficiency, and improved generalization. These results present optimal precoding as a general information-theoretic paradigm for coordinating high-dimensional actuators to control low-dimensional features.

## 1. Introduction

The human body is highly overactuated, with over 600 muscles controlling roughly 200 mechanical degrees-of-freedom. Consequently, many different combinations of muscle activations can produce the same joint torques. While this redundancy confers flexibility and robustness, it also poses 'the problem of redundancy': how does the brain choose among the infinite combinations of muscle activations that can accomplish a given movement? (Bernstein, 1967). It has long been hypothesized that the brain resolves this by controlling a small number of synergies, where each synergy recruits a group of muscles as a single functional unit (Tresch et al., 1999; d'Avella et al., 2003; Latash, 2008), effectively constraining control to a low-dimensional manifold embedded within the high-dimensional muscle space. Synergies are believed to act as fundamental building blocks for quickly learning complex new behaviors (Yang et al., 2019). They are present at birth (Sylos-Labini et al., 2020; Hinnekens et al., 2023), suggesting they encode prior knowledge about motor coordination, and are conserved across species (Dominici et al., 2011), indicating a foundational principle of biological control. Yet despite more than a century of study (Sherrington, 1910), the computational principles that underlie synergies remain poorly understood.

Understanding these principles has direct implications for the control of artificial systems: reinforcement learning (RL), the dominant paradigm for learning in high-dimensional systems, is particularly sensitive to action redundancy (Dulac-Arnold et al., 2015; Tennenholtz et al., 2019; Zahavy et al., 2018; Baram et al., 2021; Stolz et al., 2024). For value-based methods, treating behaviorally equivalent actions as distinct entities dilutes the learning signal across the action space; for policy-gradient methods, action redundancy inflates gradient variance (Wu et al., 2018). Prior approaches define action redundancy in terms of the effects of actions on the full state (Taylor et al., 2008; Rezaei-Shoshtari et al., 2022; Agarwal et al., 2020; Yang & Wang, 2020). However, in the context of musculoskeletal systems, this approach assigns equal importance to muscle and joint states—ignoring the fact that tasks are accomplished through skeletal interactions with the environment, with muscles serving merely as the actuators.

To exploit this functional asymmetry, we develop an information-theoretic framework that defines action redundancy in terms of the effects of muscle commands on joints. Our framework eliminates this redundancy by discovering a task-agnostic, state-dependent action manifold that gives an agent maximum control over its mechanical degrees-of-freedom. To quantify control, we introduce a novel information-theoretic objective: Joint-Space Empowerment (JoSE). We use this objective to optimize the action manifold, parameterized as a state-dependent precoder mapping low-dimensional latent actions to high-dimensional muscle commands (Figure 1). Through this formalization, we cast action manifold discovery as an *optimal precoding*

[1]Gatsby Unit, University College London, London, UK [2]MyoLab, New York, USA [3]King's College London, London, UK [4]Spaulding Rehab Hospital, Harvard University, Boston, USA. Correspondence to: James Heald <jamesbheald@gmail.com>.

*Proceedings of the 43rd International Conference on Machine Learning*, Seoul, South Korea. PMLR 306, 2026. Copyright 2026 by the author(s).

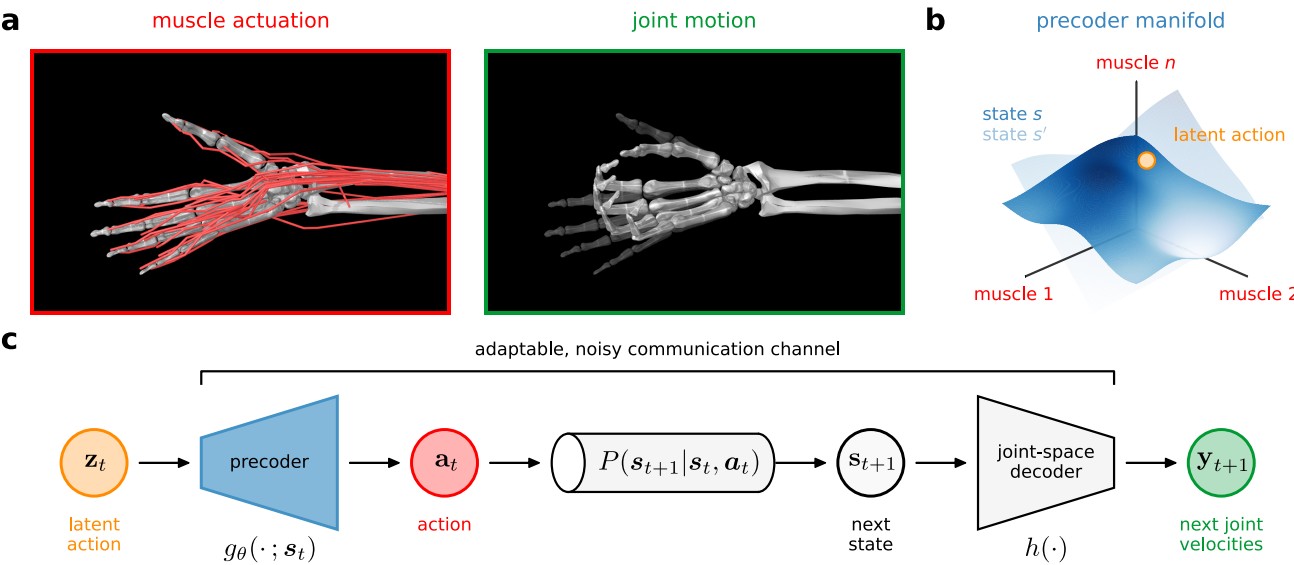

*Figure 1.* **Joint-Space Empowerment (JoSE) enables dexterous motor coordination: a,** Muscle actuation (left) produces joint motion (right) according to the dynamics of the musculoskeletal system. **b,** Each latent action (orange circle) defines a point on the low-dimensional manifold (blue) — a coordinated pattern of muscle commands. **c,** The communication problem: the sender transmits a latent action $\boldsymbol{z}_t$ across a state-dependent communication channel $p_\theta(\boldsymbol{y}_{t+1}|\boldsymbol{z}_t, \boldsymbol{s}_t)$, and the receiver observes the resultant joint velocities $\boldsymbol{y}_{t+1}$. The communication channel is composed of a state-dependent precoder $g_\theta$, the state transition kernel $P$, and a joint-space decoder $h$; the precoder transforms a low-dimensional latent action $\boldsymbol{z}_t$ into a high-dimensional action $\boldsymbol{a}_t = g_\theta(\boldsymbol{z}_t, \boldsymbol{s}_t)$ (muscle commands), the action influences the next state $\boldsymbol{s}_{t+1}$, and the joint-space decoder reads out the joint velocities $\boldsymbol{y}_{t+1} = h(\boldsymbol{s}_{t+1})$ from the next state.

*problem.* Under a control-affine Gaussian model of the musculoskeletal dynamics, we derive the optimal precoder in closed-form directly from the learned dynamics. We embed JoSE in a compositional RL algorithm, JoSEPi, in which a task-specific policy is learned on the JoSE manifold. We evaluate JoSEPi on contact-rich dexterous manipulation tasks involving the MyoHand (Caggiano et al., 2022b) (23 degrees-of-freedom, 39 muscles) and show that JoSEPi significantly improves sample efficiency, dexterity, and generalization over state-of-the-art baselines and remains robust in sparse-reward settings where task-optimized methods fail.

A precursor to JoSEPi was used to control the MyoArm (27 degrees-of-freedom, 63 muscles), achieving first place in the Manipulation Track of the NeurIPS 2024 MyoChallenge (Wang et al., 2025) and demonstrating that the method scales well beyond the MyoHand.

## 2. Related Work

### 2.1. Action Abstraction

Action redundancy occurs when multiple distinct actions produce similar effects on the environment, increasing sample complexity by artificially inflating the effective action space. Lax bisimulation metrics (Taylor et al., 2008) quantify behavioral similarity between state-action pairs based on

their immediate rewards and next-state distributions. MDP homomorphisms (Taylor et al., 2008; Rezaei-Shoshtari et al., 2022) leverage this similarity to construct abstract, lower-dimensional action spaces with theoretical guarantees on value error. The low-rank MDP literature (Agarwal et al., 2020; Yang & Wang, 2020) takes a complementary approach, exploiting low-dimensional factorizations of transition kernels for sample-efficient learning: when the transition kernel admits a low-rank factorization, sample complexity scales with the intrinsic rank rather than the ambient dimension of the action space.

### 2.2. Synergy Learning

Here we present a taxonomy of synergy learning methods.

**Demonstration-Based Approaches.** These methods extract synergies from observed movement, whether human behavior or the output of synthetic expert policies. Classical methods derive a synergy subspace by applying matrix factorization algorithms (e.g., PCA, ICA, NMF) to kinematic, muscle or motion-capture data (Santello et al., 1998; Safonova et al., 2004; Tresch et al., 2006). Deep learning has extended this approach in two directions (Song et al., 2021): variational autoencoders (VAEs) learn a nonlinear synergy manifold (Park et al., 2023; Feng et al., 2023), and

imitation learning methods distill synergies directly into a nonlinear control policy (Lee et al., 2019). These methods are descriptive: they capture coordination patterns present in the demonstration data but offer no account of how or why those patterns arise.

**Task-Optimized Approaches.** These methods derive synergies by optimizing a policy with respect to a task-specific objective function (e.g. cost-to-go or return). Optimal Feedback Control (OFC) views synergies as emergent properties of feedback control laws operating under the principle of minimum intervention (Todorov & Jordan, 2002). Deep RL methods optimize for task performance through trial-and-error interaction, embedding synergies within neural network policies rather than deriving them from analytical models of system dynamics (Chiappa et al., 2023; Wei et al., 2026). However, as demonstrated empirically, the resulting synergies remain task-specific (Chiappa et al., 2024), limiting generalization. While multi-task learning facilitates the discovery of synergies that interpolate across a family of tasks (He & Ciocarlie, 2022; Caggiano et al., 2023; Berg et al., 2023), it lacks the task-agnostic principles required for open-ended out-of-distribution extrapolation.

**Task-Agnostic Approaches.** These methods ground synergies in the biomechanics of the body, independent of task objectives. Classical methods apply control-theoretic principles to analytical dynamics models, deriving low-dimensional input subspaces that preserve the controllability of the physical system (Nori & Frezza, 2005; Berniker et al., 2009). More recent data-driven methods extract synergies from physical interaction, leveraging correlated muscle-length changes during spontaneous movement to learn a feedback control matrix (Schumacher et al., 2023) or cluster muscles into synergistic groups (He et al., 2024). However, these latter methods operate in a separate phase from task learning and lack a formal objective to maximize control of the agent's body.

### 2.3. Empowerment

The amount of control an agent can have over the world can be quantified as the capacity of the communication channel between its actions and future states, known as *empowerment* (Klyubin et al., 2005) (see Appendix A for the formal mathematical definition). Empowerment has been used for intrinsic motivation (Jung et al., 2011; Salge et al., 2014; Mohamed & Jimenez Rezende, 2015), as an objective for learning latent state representations (Bharadhwaj et al., 2022), and as a criterion for unsupervised skill discovery (Gregor et al., 2017; Eysenbach et al., 2019; Sharma et al., 2020).

## 3. Preliminaries and Problem Setting

### 3.1. Controlled Markov Process

We consider a controlled Markov process (CMP), a Markov decision process (MDP) without a reward function, defined by the tuple $\mathcal{C} = (\mathcal{S}, \mathcal{A}, P)$, where $\mathcal{S}$ and $\mathcal{A}$ are continuous state and action spaces, and $P$ is the state transition kernel. In musculoskeletal systems, each dimension of the action space represents the neural control input to a single muscle.

**Notation.** Scalars, vectors, and matrices are distinguished by font shape: scalars are non-bold, vectors are bold lower-case, and matrices are bold uppercase. Random variables are typeset in upright (roman) font, while their realizations (or fixed quantities) are typeset in italic. Thus, a random vector is bold upright (e.g., $\mathbf{x}$), and a realization of a random vector is bold italic (e.g., $\boldsymbol{x}$).

## 4. Joint-Space Empowerment (JoSE)

Our central goal is to discover a low-dimensional action manifold that maximizes how much control an agent has over its body. Rather than attempt to control the full high-dimensional state of the body, we target a task-agnostic, low-dimensional controlled variable whose regulation is sufficient to govern the body's mechanical degrees-of-freedom. In musculoskeletal systems, joint angular velocities serve this role: they are sufficient to determine body motion, while abstracting away actuator-level details such as muscle activations. Accordingly, we define the controlled variable $\boldsymbol{y}_t \in \mathcal{Y}$ as the joint angular velocities, $\boldsymbol{y}_t \triangleq \dot{\boldsymbol{q}}_t$, decoded deterministically from the full system state via a joint-space decoder $h : \mathcal{S} \to \mathcal{Y}$. Given that musculoskeletal systems are overactuated—possessing fewer joints than muscles—a low-dimensional action manifold capable of controlling joint angular velocities should exist.

We introduce a latent action space $\mathcal{Z}$ and a state-dependent precoder $g_\theta : \mathcal{Z} \times \mathcal{S} \to \mathcal{A}$, parameterized by $\theta$, which maps a latent action $\boldsymbol{z}_t \in \mathcal{Z}$ to an action $\boldsymbol{a}_t \in \mathcal{A}$. Actions produced by the precoder lie on a manifold $\mathcal{M}_\theta(\boldsymbol{s}_t) := \{g_\theta(\boldsymbol{z}_t, \boldsymbol{s}_t) : \boldsymbol{z}_t \in \mathcal{Z}\} \subseteq \mathcal{A}$ of dimensionality $\dim(\mathcal{Z})$. Both the precoder and the resulting manifold are in general state-dependent. The precoder, transition kernel, and decoder can be composed sequentially to define a conditional distribution $p_\theta(\boldsymbol{y}_{t+1}|\boldsymbol{z}_t, \boldsymbol{s}_t)$; the precoder $g_\theta$ first maps a latent action $\boldsymbol{z}_t$ to an action $\boldsymbol{a}_t$, the environment state then transitions according to $\boldsymbol{s}_{t+1} \sim P(\boldsymbol{s}_{t+1}|\boldsymbol{s}_t, \boldsymbol{a}_t)$, and finally the next state $\boldsymbol{s}_{t+1}$ is mapped to the controlled variable $\boldsymbol{y}_{t+1} = h(\boldsymbol{s}_{t+1})$ by the decoder.

We formalize our goal as an *optimal precoding problem*. Consider a communication problem involving a sender, a noisy communication channel and a receiver. At time $t$ in state $\boldsymbol{s}_t$, the sender transmits a signal $\mathbf{z}_t$ over the channel,

which, at the next time step, is received as the value $\mathbf{y}_{t+1}$. The transmitted signal, also known as the source signal, is sent to the receiver via a noisy communication channel $p_\theta(\mathbf{y}_{t+1}|\mathbf{z}_t, \mathbf{s}_t)$. The source signal $\mathbf{z}_t$ corresponds to a latent action, the communication channel is the composition of the precoder, transition kernel and decoder, and the received signal $\mathbf{y}_{t+1}$ corresponds to the resulting joint velocities.

Given a source distribution $p(\mathbf{z}_t|\mathbf{s}_t)$ and a communication channel $p_\theta(\mathbf{y}_{t+1}|\mathbf{z}_t, \mathbf{s}_t)$, the average information the received signal $\mathbf{y}_{t+1}$ contains about the transmitted signal $\mathbf{z}_t$ in state $\mathbf{s}_t$ is given by the mutual information:

$$\mathcal{I}_\theta(\mathbf{z}_t; \mathbf{y}_{t+1}|\mathbf{s}_t) = \left\langle \log \frac{p_\theta(\mathbf{y}_{t+1}, \mathbf{z}_t|\mathbf{s}_t)}{p(\mathbf{z}_t|\mathbf{s}_t)p_\theta(\mathbf{y}_{t+1}|\mathbf{s}_t)} \right\rangle_{p_\theta(\mathbf{y}_{t+1}, \mathbf{z}_t|\mathbf{s}_t)},$$

where $\langle \cdot \rangle_p$ denotes expectation with respect to distribution $p$. The greater the information transmitted by the channel, the more control or influence an agent has over the joint velocities. The mutual information can also be expressed as a difference of entropies:

$$\mathcal{I}_\theta(\mathbf{z}_t; \mathbf{y}_{t+1}|\mathbf{s}_t) = \mathcal{H}_\theta[\mathbf{y}_{t+1}|\mathbf{s}_t] - \mathcal{H}_\theta[\mathbf{y}_{t+1}|\mathbf{z}_t, \mathbf{s}_t],$$

where $\mathcal{H}_\theta[\mathbf{y}_{t+1}|\mathbf{s}_t]$ is the marginal entropy of the joint velocities, and $\mathcal{H}_\theta[\mathbf{y}_{t+1}|\mathbf{z}_t, \mathbf{s}_t]$ is the conditional entropy of the joint velocities given the latent action. The conditional entropy reflects how noisy the communication channel is. Thus, mutual information is high when a wide diversity of joint velocities can be reached (high marginal entropy) in a reliable manner (low conditional entropy). Maximizing mutual information with respect to the source distribution yields Joint-Space Empowerment (JoSE) (Definition 4.1).

**Definition 4.1** (Joint-Space Empowerment). *Given a communication channel $p_\theta(\mathbf{y}_{t+1}|\mathbf{z}_t, \mathbf{s}_t)$, the Joint-Space Empowerment (JoSE) in a state $\mathbf{s}_t \in \mathcal{S}$ is defined as the maximum mutual information between latent actions and next joint velocities over all source distributions $p(\mathbf{z}_t|\mathbf{s}_t)$ (i.e., the channel capacity):*

$$\mathcal{E}_\theta(\mathbf{s}_t) = \max_{p(\mathbf{z}_t|\mathbf{s}_t)} \mathcal{I}_\theta(\mathbf{z}_t; \mathbf{y}_{t+1}|\mathbf{s}_t).$$

Since the communication channel $p_\theta(\mathbf{y}_{t+1}|\mathbf{z}_t, \mathbf{s}_t)$ depends on the precoder $g_\theta$ whose parameters $\theta$ are learnable, the channel is itself adaptable. Our goal is to learn the optimal precoder that maximizes JoSE:

$$\theta^\star = \arg\max_\theta \mathcal{E}_\theta(\mathbf{s}_t).$$

Because the precoder $g_\theta$ defines the manifold $\mathcal{M}_\theta(\mathbf{s}_t) := \{g_\theta(\mathbf{z}_t, \mathbf{s}_t): \mathbf{z}_t \in \mathcal{Z}\} \subseteq \mathcal{A}$ that actions lie on, we frame manifold discovery as an optimal precoding problem.

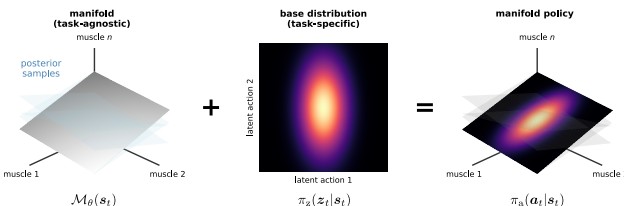

*Figure 2.* **Composing the manifold policy:** The task-specific policy $\pi_\mathrm{a}(\boldsymbol{a}_t|\boldsymbol{s}_t)$ is formed by combining two learned, state-dependent components: (1) a manifold $\mathcal{M}_\theta(\boldsymbol{s}_t)$ encoding task-agnostic muscle synergies, and (2) a base distribution $\pi_\mathrm{z}(\boldsymbol{z}_t|\boldsymbol{s}_t)$ encoding task-specific behaviors. The policy $\pi_\mathrm{a}(\boldsymbol{a}_t|\boldsymbol{s}_t)$ is the pushforward of the base distribution $\pi_\mathrm{z}(\boldsymbol{z}_t|\boldsymbol{s}_t)$ through the precoder $g_\theta$. The manifold is discovered by maximizing joint-space empowerment using a learned dynamics model (where dynamics uncertainty induces manifold uncertainty, visualized as posterior samples). The base distribution is optimized via model-free reinforcement learning. In JoSEPi, the manifold and the base distribution, can be learned simultaneously or sequentially.

## 5. Compositional Manifold Policies

We introduce a class of compositional policies that build task-specific skills on top of task-agnostic action manifolds. Specifically, a task-specific policy $\pi_\mathrm{a}(\boldsymbol{a}_t|\boldsymbol{s}_t)$ is composed of two components: a base distribution $\pi_\mathrm{z}(\boldsymbol{z}_t|\boldsymbol{s}_t)$ over latent actions, and a precoder $g_\theta(\boldsymbol{z}_t, \boldsymbol{s}_t)$ that maps low-dimensional latent actions to high-dimensional actions (subscripts on policies indicate which random variable they refer to). An action $\boldsymbol{a}_t \sim \pi_\mathrm{a}(\boldsymbol{a}_t|\boldsymbol{s}_t)$ generated by the policy is expressed as a transformation $g_\theta$ of a latent action sampled from the base distribution:

$$\boldsymbol{a}_t = g_\theta(\boldsymbol{z}_t, \boldsymbol{s}_t) \quad \text{where} \quad \boldsymbol{z}_t \sim \pi_\mathrm{z}(\boldsymbol{z}_t|\boldsymbol{s}_t).$$

Many RL algorithms incorporate the policy entropy $\mathcal{H}(\pi_\mathrm{a}(\cdot|\boldsymbol{s}_t)) = -\langle \log \pi_\mathrm{a}(\boldsymbol{a}_t|\boldsymbol{s}_t) \rangle_{\pi_\mathrm{a}}$ into the objective. For actions on the manifold $\mathcal{M}_\theta(\boldsymbol{s}_t)$, the action density $\pi_\mathrm{a}(\boldsymbol{a}_t|\boldsymbol{s}_t)$ is given by the change-of-variables formula:

$$\pi_\mathrm{a}(\boldsymbol{a}_t|\boldsymbol{s}_t) = \pi_\mathrm{z}(g_\theta^\dagger(\boldsymbol{a}_t, \boldsymbol{s}_t))$$
$$\times \left| \det\left( \boldsymbol{J}_{g_\theta}^\mathsf{T}(g_\theta^\dagger(\boldsymbol{a}_t, \boldsymbol{s}_t), \boldsymbol{s}_t) \boldsymbol{J}_{g_\theta}(g_\theta^\dagger(\boldsymbol{a}_t, \boldsymbol{s}_t), \boldsymbol{s}_t) \right) \right|^{-\frac{1}{2}},$$

where $\boldsymbol{J}_{g_\theta}(\boldsymbol{z}_t, \boldsymbol{s}_t) = \frac{\partial g_\theta(\boldsymbol{z}_t, \boldsymbol{s}_t)}{\partial \boldsymbol{z}_t}$ is the Jacobian of $g_\theta$ with respect to $\boldsymbol{z}_t$, and $g_\theta^\dagger(\cdot, \boldsymbol{s}_t): \mathcal{M}_\theta(\boldsymbol{s}_t) \to \mathcal{Z}$ is the left inverse of $g_\theta(\cdot, \boldsymbol{s}_t)$, such that $g_\theta^\dagger(g_\theta(\boldsymbol{z}_t, \boldsymbol{s}_t), \boldsymbol{s}_t) = \boldsymbol{z}_t$ for all $\boldsymbol{z}_t \in \mathcal{Z}$. The transformation $g_\theta(\cdot, \boldsymbol{s}_t)$ is assumed to be differentiable and injective, ensuring the existence of a left inverse and a well-defined density on the action manifold.

## 6. Model-Based Manifold Discovery

Estimating mutual information is notoriously difficult, especially for high-dimensional variables with complex

nonlinear dependencies (McAllester & Stratos, 2020; Belghazi et al., 2018). The difficulty stems primarily from the intractable marginal distribution $p(\boldsymbol{y}_{t+1}|\boldsymbol{s}_t) = \int p(\boldsymbol{y}_{t+1}|\boldsymbol{a}_t, \boldsymbol{s}_t)p(\boldsymbol{a}_t|\boldsymbol{s}_t) \, d\boldsymbol{a}_t$. Although variational bounds on mutual information exist (Poole et al., 2019), they suffer from the curse of dimensionality, requiring $\mathcal{O}(c^{\dim(\mathcal{A})})$ samples with $c > 1$ to estimate $p(\boldsymbol{y}_{t+1}|\boldsymbol{s}_t)$.

To develop a tractable and data-efficient solution, we adopt a model-based approach, exploiting the observation that the effects of muscle commands on joint motions can be approximated by a state-dependent control-affine mapping (see Appendix C). Specifically, we model the mapping from states and muscle commands to next joint velocities as control-affine Gaussian:

$$p(\boldsymbol{y}_{t+1}|\boldsymbol{s}_t, \boldsymbol{a}_t) = \mathcal{N}(\boldsymbol{f}(\boldsymbol{s}_t) + \boldsymbol{G}(\boldsymbol{s}_t)\boldsymbol{a}_t, \boldsymbol{Q}(\boldsymbol{s}_t)). \quad (1)$$

Crucially, while the dependence on action is affine, the dynamics parameters $\phi \triangleq \{\boldsymbol{f}, \boldsymbol{G}, \boldsymbol{Q}\}$ depend nonlinearly on the state $\boldsymbol{s}_t$. We parameterize this relationship using a neural network $\psi$ that maps the state $\boldsymbol{s}_t$ to the dynamics parameters $\phi$. We consider two variants for the input matrix $\boldsymbol{G}$: a full-rank parameterization and a low-rank factorization that reflects biomechanical coupling between joints (details in Appendix E.1.1).

For a system of the form Equation (1), an optimal precoder that maximizes JoSE can be expressed in closed form and is a function of the dynamics parameters (Theorem B.1 and Appendix B). We denote this mapping as $\boldsymbol{\theta}^* = \text{PRECODER}(\phi)$. Since $\phi$ depends on state, this yields a state-dependent matrix $\boldsymbol{\theta}^*(\boldsymbol{s}_t) \in \mathbb{R}^{\dim(\mathcal{A}) \times \dim(\mathcal{Z})}$, where $\dim(\mathcal{Z}) = \text{rank}(\boldsymbol{G})$ (Corollary B.2). The columns of this matrix span a $\dim(\mathcal{Z})$-dimensional subspace $\mathcal{M}_\theta(\boldsymbol{s}_t) := \{\boldsymbol{\theta}^*(\boldsymbol{s}_t)\boldsymbol{z}_t : \boldsymbol{z}_t \in \mathcal{Z}\} \subset \mathcal{A}$, with $\dim(\mathcal{Z}) = \dim(\mathcal{Y})$ in the full-rank case and $\dim(\mathcal{Z}) < \dim(\mathcal{Y})$ under the low-rank approximation.

For any particular state, the subspace $\mathcal{M}_\theta(\boldsymbol{s}_t)$ is locally linear and low-dimensional. However, the union of these subspaces across the state space, $\bigcup_{\boldsymbol{s}_t \in \mathcal{S}} \mathcal{M}_\theta(\boldsymbol{s}_t) \subseteq \mathcal{A}$, is not restricted to a single global linear subspace. This state-dependent parameterization allows the agent to access a nonlinear action repertoire that can span the full action space $\mathcal{A}$ over time, providing significantly greater expressivity than a fixed, state-independent subspace. In addition, estimating the dynamics parameters involves a structured regression problem that requires $\mathcal{O}(\dim(\mathcal{Y}) \cdot \dim(\mathcal{A}))$ samples—or $\mathcal{O}(\dim(\mathcal{Z})(\dim(\mathcal{Y}) + \dim(\mathcal{A})))$ when using a low-rank factorization (assuming $\boldsymbol{Q}$ is diagonal). This linear scaling is substantially more sample-efficient than variational approaches, enabling scalable precoder identification.

## 7. Epistemic Manifold Uncertainty

The closed-form construction in Theorem B.1 assumes that the dynamics parameters are known. However, in practice, these parameters must be learned from data. This creates a fundamental conflict: learning a globally accurate dynamics model requires exploring the full action space $\mathcal{A}$, yet the compositional policy $\pi_a(\boldsymbol{a}_t|\boldsymbol{s}_t)$ confines actions to a low-dimensional manifold.

To resolve this, we represent uncertainty about the dynamics parameters in the form of a distribution $p(\phi|\boldsymbol{s}_t)$. This induces an implicit distribution over the optimal precoder:

$$p(\boldsymbol{\theta}^*|\boldsymbol{s}_t) = \langle \delta\left(\boldsymbol{\theta}^* - \text{PRECODER}(\phi)\right)\rangle_{p(\phi|\boldsymbol{s}_t)}, \quad (2)$$

where $\delta(\cdot)$ denotes the Dirac delta function. A sample from the optimal precoder distribution can be obtained by drawing a sample $\hat{\phi} \sim p(\phi|\boldsymbol{s}_t)$ and then evaluating $\text{PRECODER}(\hat{\phi})$.

Early in learning, high uncertainty in $p(\phi|\boldsymbol{s}_t)$ produces a broad distribution over the optimal precoder $p(\boldsymbol{\theta}^*|\boldsymbol{s}_t)$, enabling the agent to explore effectively across the entire action space $\mathcal{A}$. As dynamics uncertainty diminishes during learning, the induced distribution progressively concentrates around a consistent and stable manifold. The result is a natural, uncertainty-driven transition from wide-ranging exploration to precise, synergistic control aligned with the learned dynamics.

This procedure is closely related to Thompson sampling (Thompson, 1933) in that exploration arises from sampling latent model parameters from a posterior and acting optimally under the sampled model.

## 8. Reinforcement Learning

To evaluate the proposed framework, we train musculoskeletal agents to perform dexterous manipulation. Dexterous manipulation with a human hand serves as an ideal testbed for evaluating our framework. The human hand poses a challenging control problem, requiring the coordination of many redundant muscles—both complementary and antagonistic—to produce precise forces amid intermittent, contact-rich dynamics.

To capture these complexities, we utilize the physiologically accurate MyoHand model of the human hand. This musculoskeletal model is implemented in the MuJoCo physics simulator and features 29 bones, 23 joints and 39 muscle-tendon units, resulting in high-dimensional, overactuated, and nonlinear dynamics. We evaluate our framework on a suite of tasks that require in-hand reorientation or repositioning of one or more objects, necessitating highly synchronized control of the forearm, wrist and finger muscles.

Each task is formulated as a Markov decision process (MDP) that can be solved via RL (Sutton & Barto, 2018). An MDP

**Algorithm 1** JoSEPi

**Input:** initial base distribution $\pi_z$, initial (or pretrained) dynamics network $\psi$, empty buffer $\mathcal{B}$
**for** each iteration **do**
    **for** each environment step **do**
        $z_t \sim \pi_z(z_t|s_t)$                 *sample latent action*
        $\hat{\phi} \sim q_\psi(\phi|s_t, \mathcal{D})$        *sample dynamics*
        $\theta \leftarrow \text{PRECODER}(\hat{\phi})$      *compute precoder*
        $a_t = \theta z_t$               *generate action*
        $s_{t+1} \sim P(s_{t+1}|s_t, a_t)$    *step environment*
        $\mathcal{B} \leftarrow \mathcal{B} \cup \{s_t, a_t, r_t, s_{t+1}\}$   *store transition*
    **end for**
    **for** each gradient step **do**
        train $\psi$ via MLE      *update dynamics model*
        train $\pi_z$ with RL    *update base distribution*
    **end for**
**end for**

$(\mathcal{S}, \mathcal{A}, T, r, \rho_0)$ is a CMP with an additional reward function $r$ and initial state distribution $\rho_0$. The goal in RL is to learn a policy $\pi(a_t|s_t)$ that maximizes the expected return $\sum_t^T \langle r(s_t, a_t) \rangle_{\rho_\pi}$, where the expectation is taken with respect to the state-action trajectory distribution $\rho_\pi$ induced by the policy.

### 8.1. JoSEPi

We train the compositional policy $\pi_a(a_t|s_t)$ using a combination of model-based and model-free updates. The base distribution $\pi_z(z_t|s_t)$ is optimized using soft actor-critic (SAC), an off-policy actor-critic algorithm based on the maximum entropy RL framework. Gradients are propagated through the precoder, allowing the base distribution to adapt while respecting the structure of the low-dimensional action manifold.

The dynamics network $\psi$ is trained via maximum likelihood estimation on a dataset of transitions $\mathcal{D} = \{(s_t^{(i)}, a_t^{(i)}, y_{t+1}^{(i)})\}_{i=1}^n$ (e.g., a replay buffer). We apply Monte Carlo (MC) dropout to the hidden units (Gal & Ghahramani, 2016), inducing an implicit variational posterior $q_\psi(\phi|s_t, \mathcal{D})$ over the state-dependent dynamics parameters. For the low-rank approximation to the input matrix $G$, we use rank 15, yielding a 15-dimensional state-dependent manifold $\mathcal{M}_\theta(s_t)$ in a 39-dimensional action space $\mathcal{A}$. Unless stated otherwise, we report results for the low-rank variant.

Training proceeds iteratively: we use the current policy to interact with the environment, store the resulting transitions in a replay buffer $\mathcal{B}$, and then perform gradient updates on both the dynamics network $\psi$ and the base distribution $\pi_z(z_t|s_t)$. Whenever an action $a_t \sim \pi_a(a_t|s_t)$ is required, we first draw a sample realization of the dynamics

$\hat{\phi} \sim q_\psi(\phi|s_t, \mathcal{D})$ (via Monte Carlo dropout), and then obtain $\theta = \text{PRECODER}(\hat{\phi})$. The full procedure is summarized in Algorithm 1. We refer to our algorithm—which combines the JoSE objective with an RL algorithm for learning a policy ($\pi$)—as JoSEPi. Hyperparameters and implementation details are summarized in Appendix E.

## 9. Experiments

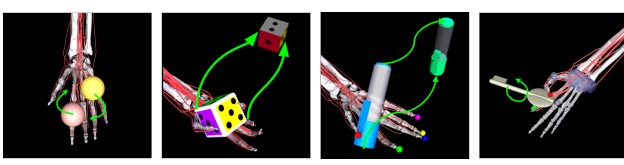

*Figure 3.* **In-hand object manipulation with the MyoHand:** We evaluate JoSEPi on a suite of contact-rich object manipulation tasks from MyoSuite. From left to right: BaodingBalls, DieReorient, PenTwirl, and KeyTurn.

We evaluate JoSEPi on a suite of contact-rich in-hand manipulation tasks from MyoSuite (Figure 3) (Caggiano et al., 2022b), including the two tasks featured in the NeurIPS MyoChallenge 2022 competition (Caggiano et al., 2022a): BaodingBalls and DieReorient. In **BaodingBalls**, the MyoHand must simultaneously rotate two free-floating balls over the palm, requiring precise coordination across multiple fingers. In **DieReorient**, the MyoHand must reconfigure a die to match target orientations, demanding delicate control to manipulate the object without dropping it. In **KeyTurn**, the MyoHand must rotate a key, primarily using the index finger and thumb, with intermittent contact throughout the task. In **PenTwirl**, the MyoHand must rotate a pen to a target orientation while maintaining stability through intermittent multi-finger contacts.

In the **Reorient** family of environments, the MyoHand rotates objects drawn from four geometry classes (ellipsoid, box, capsule, sphere), with either two (**Reorient8**) or twenty-five (**Reorient100**) distinct geometric configurations, toward predetermined target orientations. After training on Reorient100, zero-shot generalization is evaluated on 1000 novel objects with in-domain (**ReorientID**) or out-of-domain (**ReorientOOD**) geometric configurations.

In addition to these benchmark tasks, we evaluate a custom sparse-reward variant of Reorient8 to study the effect of reward sparsity on manifold discovery and task learning. In this setting, the agent receives a reward of 1 in solved states and 0 otherwise, using the default Reorient8 solved criterion.

Following prior work, we evaluate model performance using a *fraction solved* metric, defined as the ratio between the

number of time steps considered solved within an episode and the maximum length of the episode. This is generally less than 1.0 (even for the optimal policy), as it takes time to reach solved states.

## 9.1. Baselines

We compare JoSEPi with current model-free RL methods in overactuated systems: SAC (Haarnoja et al., 2018), Lattice (Chiappa et al., 2023), Synergistic Action Representation (SAR) (Berg et al., 2023), and a state-dependent variant of SAR (SD-SAR) (Appendix D.2.2). Like JoSEPi, SAR and SD-SAR are integrated with SAC. Lattice was originally integrated with both SAC and PPO. In the case of PPO, recurrent LSTM actor and critic networks were used to address partial observability (e.g. unobserved object properties like friction, mass, shape). We implement both versions and denote them Lattice-SAC and Lattice-rPPO.

For fair comparison, all SAC-based methods (JoSEPi, Lattice-SAC, SAR, SD-SAR, SAC) use the same standard hyperparameters and network architecture (Table 2). For Lattice-rPPO, we use the hyperparameters published in Chiappa et al. (2023). The same hyperparameters are used for all experiments. Following Haarnoja et al. (2018), actions are bounded via a tanh nonlinearity, with the change-of-variables formula applied to compute log-probabilities for entropy estimation. All results are averaged across 5 random seeds.

# 10. Results

## 10.1. Simultaneous Manifold and Task Learning

We first evaluate JoSEPi in the most demanding setting, where both the state-dependent action manifold $\mathcal{M}_\theta(s_t)$ and the task-specific base distribution $\pi_z(z_t|s_t)$ are learned simultaneously. As shown in Figure 4, JoSEPi consistently outperforms the baselines across all tasks, achieving faster learning, higher asymptotic performance, and improved training stability—confirming that a structured, low-dimensional action manifold provides a strong inductive bias for exploration and control in overactuated systems. On KeyTurn, two baselines approach JoSEPi's asymptotic performance, though JoSEPi still converges approximately 2–3× faster.

To investigate the mechanisms underpinning these performance gains, we systematically evaluate the properties of the learned dynamics model. A post-hoc analysis confirms that our control-affine approximation maintains high prediction accuracy even during local discontinuities caused by contact events (Section F.1). The MC dropout ensemble also provides meaningful estimates of epistemic uncertainty (Section F.2), and an ablation confirms that this uncertainty is crucial for training stability and asymptotic task perfor-

mance (Section G.1). Furthermore, JoSEPi is robust to the dimensionality of the action manifold, which is determined by the input matrix factorization rank (Section G.2). Interestingly, a traditional matrix factorization algorithm applied to the muscle activations overestimates the dimensionality of control, as it ignores the state-dependent nature of the action manifold (Section H).

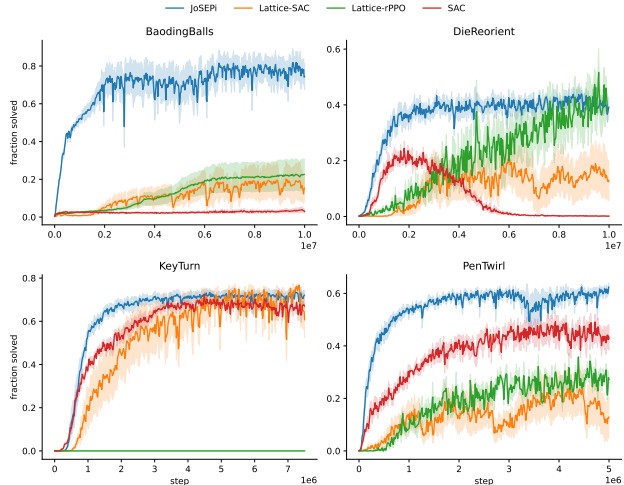

*Figure 4.* **Simultaneous manifold and task learning:** Performance of JoSEPi and baselines on BaodingBalls, DieReorient, KeyTurn, and PenTwirl. Data show solved fraction over training (mean ± s.e.m. across 5 random seeds). JoSEPi learns the task-agnostic manifold and the task-specific base distribution simultaneously.

## 10.2. Dynamics-Based Manifold Discovery

A key feature of JoSEPi is that the action manifold is derived from a learned dynamics model. Because every state transition provides a data point for training the dynamics model, the learning signal for manifold discovery is dense. In contrast, task-optimized approaches learn synergies from extrinsic reward signals, which are not guaranteed to be dense. For example, Lattice learns a full-covariance policy through model-free RL, and so may struggle to achieve meaningful actuator correlations if rewards are infrequent.

Sparse-reward environments provide a direct stress test of this distinction, isolating whether effective actuator coordination can emerge with minimal extrinsic supervision. Such settings also offer a simpler way to specify manipulation goals, avoiding hand-crafted reward functions that can lead to unintended behaviors (reward hacking).

As shown in Figure 5, JoSEPi remains robust under sparse rewards, substantially outperforming all baselines in both learning speed and final task performance. This confirms

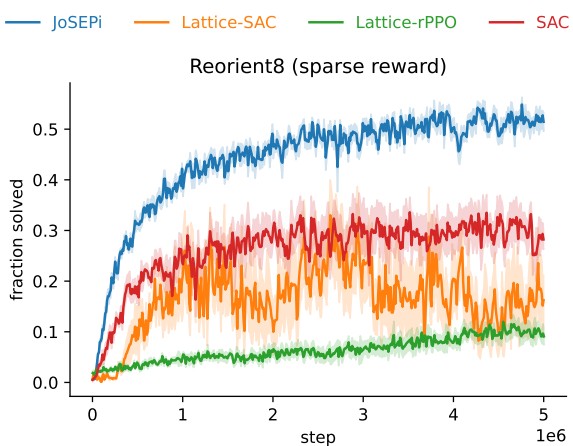

*Figure 5.* **JoSEPi discovers manifolds from dynamics, not rewards:** Performance of JoSEPi and baselines on a sparse-reward variant of Reorient8. Data show solved fraction over training (mean ± s.e.m. across 5 random seeds). JoSEPi maintains strong performance under reward sparsity due to its model-based approach to manifold discovery.

that dynamics-based manifold discovery enables effective coordination even when reward signals are rare.

### 10.3. Decoupled Manifold and Task Learning

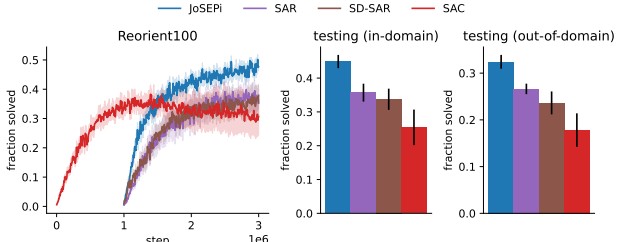

*Figure 6.* **Decoupled manifold and task learning:** Performance of JoSEPi and baselines on Reorient100 (left) and zero-shot generalization to unseen objects (middle, right). Data show solved fraction (mean ± s.e.m. across 5 random seeds). JoSEPi and SAR are offset by 1M steps to account for their play phase, whereas SAC learns from scratch.

In JoSEPi, the policy factorizes into two components: a task-agnostic manifold $\mathcal{M}_\theta(s_t)$, optimized with respect to joint-space empowerment, and a task-specific base distribution $\pi_z(z_t|s_t)$, optimized with respect to the RL objective. Once the task-agnostic manifold is acquired, learning a new task reduces to optimizing $\pi_z(z_t|s_t)$ with the manifold held fixed. This shifts the policy search from the high-

dimensional action space $\mathcal{A}$ to the low-dimensional latent space $\mathcal{Z}$. We exploit this compositional structure by first discovering a manifold in a play phase, and then using the manifold to accelerate learning on a downstream task.

In the play phase, a SAC agent is trained on a relatively simple task (Reorient8). The resulting state transitions are used to train the dynamics model, from which the JoSE-optimized action manifold is derived. We compare JoSEPi against two demonstration-based approaches to learning synergies. SAR identifies a *state-independent* subspace by applying ICA and PCA (ICAPCA) to muscle activations from the play-phase policy. SD-SAR identifies a *state-dependent* subspace by training a state-conditioned linear autoencoder using the same state information available to JoSEPi, but replacing the JoSE objective with a reconstruction loss. We include this baseline to disentangle the contribution of state-dependent representational capacity—which SAR lacks—from the contribution of the JoSE objective. In all cases, the manifold is held fixed as a more difficult downstream task (Reorient100) is learned.

As shown in Figure 6 (left), JoSEPi substantially outperforms SAR, SD-SAR, and SAC on the downstream task. The advantage of JoSEPi over SD-SAR—a baseline that shares its state-dependent parameterization but optimizes reconstruction error—confirms that the performance gains are driven by the JoSE objective, not by representational capacity alone. This advantage arises because demonstration-based methods are inherently task-specific: by capturing coordination patterns present in the play-phase data, they extract a manifold that reflects specific play-phase behaviors, limiting transferability to new tasks. The task-agnostic JoSE objective, in contrast, yields a dynamics-derived manifold that transfers robustly as a reusable foundation for building new policies.

After training on Reorient100, we evaluate zero-shot generalization of the trained policy to 1000 unseen objects with in-domain (ReorientID) and out-of-domain (ReorientOOD) properties. JoSEPi achieves the best performance in both conditions (Figure 6, right), demonstrating that the JoSE-optimized manifold provides an effective scaffold for learning policies that generalize more robustly to novel objects and geometries.

## 11. Discussion

In this work, we formalize muscle synergies as an optimal precoder that maximizes information transmission over a noisy communication channel. The precoding framework underpinning JoSE has deep roots in communication theory. In classical multi-input multi-output (MIMO) systems, a precoder transforms a source signal before it is transmitted across the channel to maximize the mutual information be-

tween the transmitted and received signals (Palomar et al., 2003). Under a linear-Gaussian channel model, the optimal precoder aligns the transmitted signal with the right singular vectors of the channel matrix and allocates power across those directions using water-filling (Telatar, 1999). Our theory establishes a direct analogy between this classical model and motor coordination: the motor system is the source, the muscle synergies are the precoder, the musculoskeletal system is the noisy channel, and the joint kinematics are the received signal.

Our method is a task-agnostic approach to learning synergies (Section 2.2), as JoSE is fully specified by the dynamics of the system; it operates within a Controlled Markov Process (CMP) that has no reward function. This distinguishes it from task-optimized approaches such as Optimal Feedback Control (OFC) (Todorov & Jordan, 2002), where synergies are inherently task-specific and must be re-computed for each new task. Consequently, OFC lacks a mechanism for transfer learning across tasks (though extensions exist that mix task-specific control laws (Todorov, 2009)). In contrast, JoSEPi enables transfer by sharing task-agnostic action manifolds across task-specific compositional policies. Our method is also distinct from control-theoretic approaches that assume deterministic dynamics (Nori & Frezza, 2005; Berniker et al., 2009). By using an information-theoretic formulation of control, JoSE takes into account the noise properties of the musculoskeletal system. As a result, it can identify synergy manifolds that are robust to action-dependent transition stochasticity ignored by prior methods.

Task-optimized methods like OFC produce synergies specialized for a single task, while task-agnostic methods like JoSE produce synergies that generalize across tasks by targeting the full mechanical degrees-of-freedom of the body. To harness the benefits of both specialization and generalization, the controlled variable could be treated as a free design choice, yielding a more general objective for learning action manifolds—Control-Space Empowerment (CoSE). By targeting a controlled variable that can be shared across a family of related tasks, CoSE could learn synergies that balance specialization and generalization. The general strategy of controlling specific state features rather than the entire state is a core tenet of human motor control, articulated in the uncontrolled manifold hypothesis (Scholz et al., 2000) and the minimum intervention principle (Todorov & Jordan, 2002). CoSE translates this biological principle into a formal computational objective, with the control-space decoder specifying which state feature to control. A decoder targeting object velocity, for instance, could yield action manifolds capable of supporting a broad family of dexterous manipulation tasks; and a decoder targeting center-of-mass velocity could yield action manifolds that support a family of locomotion behaviors, from walking to running and jumping. Together, control-space decoding and optimal

precoding offer a general, information-theoretic framework for coordinating high-dimensional actuators to control low-dimensional features of the body and world.

**Limitations and Future Work**    The JoSEPi algorithm introduces two key approximations to maintain analytical tractability. First, we model the dynamics as control-affine, which cannot capture action-dependent nonlinearities (e.g., contact discontinuities caused by varying actions in a fixed state). Second, we assume additive channel noise, whereas physiological motor noise is signal-dependent (Harris & Wolpert, 1998).

The core theoretical framework established here invites several directions for future work. For instance, our one-step empowerment objective (i.e., mutual information between current latent actions and joint velocities at the next time step) could be extended to multiple time steps, enabling the discovery of spatiotemporal synergies that mediate influence over controlled variables farther in the future. Additionally, our representation of dynamics uncertainty could be used to drive intrinsically motivated exploration (Sun et al., 2011; Houthooft et al., 2016), allowing manifold discovery to remain entirely within an unsupervised, information-theoretic framework. Another important direction is to deploy JoSEPi in physical robotic platforms, which will necessitate addressing sim-to-real challenges (e.g., unmodeled tendon friction and complex sensor noise). Finally, as the primary focus of this work is to establish a theoretical formalism, we leave direct empirical comparisons between our learned synergies and biological data for future work.

## Software and Data

The source code to reproduce the experiments is publicly available at https://github.com/gatsby-sahani/JoSE.

## Acknowledgements

We would like to thank the anonymous reviewers for their valuable feedback, which helped improve the quality of this manuscript. We also express our gratitude to our colleagues for insightful discussions, with a special thanks to Benjamin Russell for his input regarding subspace bases.

## Impact Statement

This work is aimed at advancing motor coordination in high-dimensional systems with potential downstream benefits for robotics (Ficuciello et al., 2019; Stella et al., 2025), prosthetics (Capsi-Morales et al., 2025), rehabilitation (Safavynia et al., 2011), and wearable neural interfaces (Kaifosh & Reardon, 2025).

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

# A. Empowerment

Empowerment has been proposed as an intrinsic objective for driving autonomous behavior in biological and artificial agents (Klyubin et al., 2005). Formally, it is defined as the capacity of the communication channel between actions and next states (Definition A.1).

**Definition A.1** (Empowerment). *Given a communication channel $p(\boldsymbol{s}_{t+1}|\boldsymbol{s}_t, \boldsymbol{a}_t)$, the empowerment $\mathcal{E}(\boldsymbol{s}_t)$ in a state $\boldsymbol{s}_t \in \mathcal{S}$ is defined as the maximum mutual information between actions and next states over all source distributions $p(\boldsymbol{a}_t|\boldsymbol{s}_t)$ (i.e., the channel capacity):*

$$\mathcal{E}(\boldsymbol{s}_t) = \max_{p(\boldsymbol{a}_t|\boldsymbol{s}_t)} \mathcal{I}(\mathbf{a}_t; \mathbf{s}_{t+1}|\boldsymbol{s}_t).$$

# B. The Optimal Precoding Problem

## B.1. The Setting

Consider a communication problem involving a sender, a receiver and a communication channel. The sender transmits a signal drawn from a Gaussian source distribution $p(\boldsymbol{z}_t|\boldsymbol{s}_t) = \mathcal{N}(\mathbf{0}, \boldsymbol{\Sigma}(\boldsymbol{s}_t))$. The source signal is sent to the receiver via a control-affine communication channel $p(\boldsymbol{y}_{t+1}|\boldsymbol{z}_t, \boldsymbol{s}_t)$:

$$\boldsymbol{y}_{t+1} = \boldsymbol{f}(\boldsymbol{s}_t) + \boldsymbol{G}(\boldsymbol{s}_t)\boldsymbol{\theta}(\boldsymbol{s}_t)\boldsymbol{z}_t + \boldsymbol{\epsilon}_t, \quad \boldsymbol{\epsilon}_t \sim \mathcal{N}(\mathbf{0}, \boldsymbol{Q}(\boldsymbol{s}_t)).$$

Here $\boldsymbol{G} \in \mathbb{R}^{\dim(\mathcal{Y}) \times \dim(\mathcal{A})}$, $\boldsymbol{\theta} \in \mathbb{R}^{\dim(\mathcal{A}) \times \dim(\mathcal{Z})}$, $\boldsymbol{z}_t \in \mathbb{R}^{\dim(\mathcal{Z})}$, and $\boldsymbol{Q} \in \mathbb{R}^{\dim(\mathcal{Y}) \times \dim(\mathcal{Y})}$.

The optimal precoding problem is to find the precoder $\boldsymbol{\theta}(\boldsymbol{s}_t)$ that maximizes the capacity of the communication channel.

For the remainder of the section, we suppress the dependence of the parameters $\boldsymbol{f}$, $\boldsymbol{G}$, $\boldsymbol{\theta}$, $\boldsymbol{Q}$ and $\boldsymbol{\Sigma}$ on the state $\boldsymbol{s}_t$ to simplify notation.

## B.2. The Problem

Maximizing the capacity of the communication channel with respect to the precoder is equivalent to maximizing the mutual information with respect to both the precoder and the covariance of the source distribution:

$$\begin{aligned} \underset{\boldsymbol{\theta}, \boldsymbol{\Sigma}}{\text{maximize}} \quad & \mathcal{I}_{\theta}(\mathbf{z}_t; \mathbf{y}_{t+1}|\boldsymbol{s}_t) \\ \text{subject to} \quad & \boldsymbol{\Sigma} \succeq 0, \ \text{Tr}(\boldsymbol{\theta}\boldsymbol{\Sigma}\boldsymbol{\theta}^{\mathsf{T}}) = P, \end{aligned}$$
(P1a)

where $\boldsymbol{\Sigma} \succeq 0$ denotes that $\boldsymbol{\Sigma}$ is positive semidefinite (PSD). The constraint on the total transmit power $\text{Tr}(\boldsymbol{\theta}\boldsymbol{\Sigma}\boldsymbol{\theta}^{\mathsf{T}})$ (the trace of the covariance of the transmitted signal) ensures that the mutual information is bounded; the capacity of an affine–Gaussian channel is infinite otherwise.

Because the channel noise is independent of the source signal, maximizing the mutual information $\mathcal{I}_{\theta}(\mathbf{z}_t; \mathbf{y}_{t+1}|\boldsymbol{s}_t) = \mathcal{H}_{\theta}[\mathbf{y}_{t+1}|\boldsymbol{s}_t] - \mathcal{H}_{\theta}[\mathbf{y}_{t+1}|\mathbf{z}_t, \boldsymbol{s}_t]$ is equivalent to maximizing

the marginal entropy $\mathcal{H}_\theta[\mathbf{y}_{t+1}|\mathbf{s}_t]$. As the marginal distribution for the setting considered here is $p_\theta(\mathbf{y}_{t+1}|\mathbf{s}_t) = \mathcal{N}(\mathbf{f}, \mathbf{G}\boldsymbol{\theta}\boldsymbol{\Sigma}\boldsymbol{\theta}^\mathsf{T}\mathbf{G}^\mathsf{T} + \mathbf{Q})$, the optimization problem in P1a can be equivalently written as

$$\begin{array}{ll} \underset{\boldsymbol{\theta},\boldsymbol{\Sigma}}{\text{maximize}} & \det(\mathbf{G}\boldsymbol{\theta}\boldsymbol{\Sigma}\boldsymbol{\theta}^\mathsf{T}\mathbf{G}^\mathsf{T} + \mathbf{Q}) \\ \text{subject to} & \boldsymbol{\Sigma} \succeq 0, \ \text{Tr}(\boldsymbol{\theta}\boldsymbol{\Sigma}\boldsymbol{\theta}^\mathsf{T}) = P. \end{array} \quad \text{(P1b)}$$

**B.3. The Solution**

Let $\mathbf{Q} = \boldsymbol{\Psi}\boldsymbol{\Lambda}\boldsymbol{\Psi}^\mathsf{T}$ denote the eigendecomposition of the noise covariance matrix $\mathbf{Q}$, and define the whitened channel matrix $\tilde{\mathbf{G}} = \boldsymbol{\Lambda}^{-\frac{1}{2}}\boldsymbol{\Psi}^\mathsf{T}\mathbf{G}$. Since $\det(\mathbf{G}\boldsymbol{\theta}\boldsymbol{\Sigma}\boldsymbol{\theta}^\mathsf{T}\mathbf{G}^\mathsf{T} + \mathbf{Q}) = \det(\mathbf{Q})\det(\tilde{\mathbf{G}}\boldsymbol{\theta}\boldsymbol{\Sigma}\boldsymbol{\theta}^\mathsf{T}\tilde{\mathbf{G}}^\mathsf{T} + \mathbf{I})$ and $\det(\mathbf{Q})$ does not depend on $\boldsymbol{\theta}$ or $\boldsymbol{\Sigma}$, the argmax of P1b coincides with that of

$$\begin{array}{ll} \underset{\boldsymbol{\theta},\boldsymbol{\Sigma}}{\text{maximize}} & \det(\tilde{\mathbf{G}}\boldsymbol{\theta}\boldsymbol{\Sigma}\boldsymbol{\theta}^\mathsf{T}\tilde{\mathbf{G}}^\mathsf{T} + \mathbf{I}) \\ \text{subject to} & \boldsymbol{\Sigma} \succeq 0, \ \text{Tr}(\boldsymbol{\theta}\boldsymbol{\Sigma}\boldsymbol{\theta}^\mathsf{T}) = P, \end{array} \quad \text{(P1c)}$$

where $\tilde{\mathbf{G}}\boldsymbol{\theta}\boldsymbol{\Sigma}\boldsymbol{\theta}^\mathsf{T}\tilde{\mathbf{G}}^\mathsf{T} + \mathbf{I}$ denotes the marginal covariance expressed in the whitened basis.

Next, let $\tilde{\mathbf{G}} = \mathbf{U}_r\mathbf{S}_r\mathbf{V}_r^\mathsf{T}$ denote the compact singular value decomposition (SVD) of the rank-$r$ matrix $\tilde{\mathbf{G}}$, retaining only nonzero singular values and their corresponding singular vectors; thus $\mathbf{U}_r \in \mathbb{R}^{\dim(\mathcal{Y}) \times r}$, $\mathbf{S}_r \in \mathbb{R}^{r \times r}$, and $\mathbf{V}_r \in \mathbb{R}^{\dim(\mathcal{A}) \times r}$.

**Theorem B.1** (Optimal Precoder)**.** *A solution to the optimization problem in* P1a *is given by*

$$\boldsymbol{\theta}^* = \mathbf{V}_r,$$

*the right singular vectors of* $\tilde{\mathbf{G}}$ *associated with nonzero singular values.*

*Proof.* The objective depends on $\boldsymbol{\theta}$ and $\boldsymbol{\Sigma}$ only through the covariance of the transmitted signal $\boldsymbol{\Phi} \triangleq \boldsymbol{\theta}\boldsymbol{\Sigma}\boldsymbol{\theta}^\mathsf{T}$, so problem P1c is equivalent to

$$\begin{array}{ll} \underset{\boldsymbol{\Phi}}{\text{maximize}} & \det(\tilde{\mathbf{G}}\,\boldsymbol{\Phi}\,\tilde{\mathbf{G}}^\mathsf{T} + \mathbf{I}) \\ \text{subject to} & \boldsymbol{\Phi} \succeq 0, \ \text{Tr}(\boldsymbol{\Phi}) = P. \end{array} \quad \text{(P1d)}$$

Any component of $\boldsymbol{\Phi}$ that lies in the nullspace of $\tilde{\mathbf{G}}$ makes zero contribution to $\tilde{\mathbf{G}}\boldsymbol{\Phi}\tilde{\mathbf{G}}^\mathsf{T}$. Since $\det(\tilde{\mathbf{G}}\boldsymbol{\Phi}\tilde{\mathbf{G}}^\mathsf{T} + \mathbf{I})$ is monotone non-decreasing on the PSD cone (Boyd & Vandenberghe, 2004), redirecting power from the nullspace into $\text{range}(\mathbf{V}_r)$ — its orthogonal complement under the compact SVD $\tilde{\mathbf{G}} = \mathbf{U}_r\mathbf{S}_r\mathbf{V}_r^\mathsf{T}$ — cannot decrease the objective. An optimal solution can therefore be parameterized as $\boldsymbol{\Phi} = \mathbf{V}_r\boldsymbol{\Gamma}\mathbf{V}_r^\mathsf{T}$ for some $\boldsymbol{\Gamma} \succeq 0$, where the power constraint reduces to $\text{Tr}(\boldsymbol{\Gamma}) = P$ by the cyclic property of the trace.

Substituting $\tilde{\mathbf{G}} = \mathbf{U}_r\mathbf{S}_r\mathbf{V}_r^\mathsf{T}$, $\boldsymbol{\Phi} = \mathbf{V}_r\boldsymbol{\Gamma}\mathbf{V}_r^\mathsf{T}$ and applying $\det(\mathbf{I} + \mathbf{A}\mathbf{B}) = \det(\mathbf{I} + \mathbf{B}\mathbf{A})$ gives

$$\det(\tilde{\mathbf{G}}\,\boldsymbol{\Phi}\,\tilde{\mathbf{G}}^\mathsf{T} + \mathbf{I}) = \det(\mathbf{S}_r\boldsymbol{\Gamma}\mathbf{S}_r^\mathsf{T} + \mathbf{I}),$$

and hence P1d reduces to

$$\begin{array}{ll} \underset{\boldsymbol{\Gamma}}{\text{maximize}} & \det(\mathbf{S}_r\boldsymbol{\Gamma}\mathbf{S}_r^\mathsf{T} + \mathbf{I}) \\ \text{subject to} & \boldsymbol{\Gamma} \succeq 0, \ \text{Tr}(\boldsymbol{\Gamma}) = P. \end{array} \quad \text{(P1e)}$$

By the Hadamard inequality, for any positive semidefinite matrix $\mathbf{K}$,

$$\det(\mathbf{K}) \leq \prod_i [\mathbf{K}]_{ii},$$

with equality iff $\mathbf{K}$ is diagonal. Applied to $\mathbf{K} = \mathbf{S}_r\boldsymbol{\Gamma}\mathbf{S}_r^\mathsf{T} + \mathbf{I}$, the objective is bounded by $\prod_i(\sigma_i^2[\boldsymbol{\Gamma}]_{ii}+1)$, where $\sigma_i$ are the singular values in $\mathbf{S}_r$. This bound depends only on the diagonal of $\boldsymbol{\Gamma}$, which suggests discarding the off-diagonal entries. To this end, let $\boldsymbol{\Gamma}_\mathrm{d}$ denote the matrix obtained by setting the off-diagonal entries of $\boldsymbol{\Gamma}$ to zero. Since both $\boldsymbol{\Gamma}_\mathrm{d}$ and $\mathbf{S}_r$ are diagonal, $\mathbf{K} = \mathbf{S}_r\boldsymbol{\Gamma}_\mathrm{d}\mathbf{S}_r^\mathsf{T} + \mathbf{I}$ is diagonal, and hence by the equality condition above, $\boldsymbol{\Gamma}_\mathrm{d}$ attains the bound:

$$\det(\mathbf{S}_r\boldsymbol{\Gamma}\mathbf{S}_r^\mathsf{T}+\mathbf{I}) \leq \prod_i(\sigma_i^2[\boldsymbol{\Gamma}]_{ii}+1) = \det(\mathbf{S}_r\boldsymbol{\Gamma}_\mathrm{d}\mathbf{S}_r^\mathsf{T}+\mathbf{I}).$$

Therefore, replacing any feasible $\boldsymbol{\Gamma}$ by $\boldsymbol{\Gamma}_\mathrm{d}$ does not decrease the objective, so an optimal solution $\boldsymbol{\Gamma}^*$ to P1e can be taken to be diagonal. Recalling that $\boldsymbol{\Phi} = \mathbf{V}_r\boldsymbol{\Gamma}\mathbf{V}_r^\mathsf{T}$, an optimal transmit covariance for P1d is $\boldsymbol{\Phi}^* = \mathbf{V}_r\boldsymbol{\Gamma}^*\mathbf{V}_r^\mathsf{T}$, which is realized by the precoder $\boldsymbol{\theta}^* = \mathbf{V}_r$ together with a diagonal source covariance $\boldsymbol{\Sigma}^* = \boldsymbol{\Gamma}^*$. This proves that $\mathbf{V}_r$, the right singular vectors of $\tilde{\mathbf{G}}$ associated with nonzero singular values, is an optimal precoder.

$\square$

**Corollary B.2** (Optimal Number of Synergies)**.** *The closed-form solution* $\boldsymbol{\theta}^* = \mathbf{V}_r \in \mathbb{R}^{\dim(\mathcal{A}) \times r}$ *enforces* $\dim(\mathcal{Z}) = r = \text{rank}(\tilde{\mathbf{G}})$. *Each column of* $\mathbf{V}_r$ *corresponds to a single synergy, and the* $r$ *synergies span an* $r$-*dimensional subspace of the action space* $\mathcal{A}$. *Any synergy orthogonal to this subspace lies in the nullspace of* $\tilde{\mathbf{G}}$, *contributes nothing to the objective, and receives zero power at optimality. For overactuated musculoskeletal systems, where muscles outnumber joints* $(\dim(\mathcal{A}) > \dim(\mathcal{Y}))$, *the full-rank case gives* $r = \dim(\mathcal{Y})$, *that is, the number of synergies equals the number of joints.*

*Remark* B.3 (Non-Uniqueness)**.** The optimal precoder is not unique. Any precoder whose columns span the same subspace as $\mathbf{V}_r$ is empowerment-optimal. Formally, for any invertible matrix $\mathbf{T}$, the precoder $\mathbf{V}_r\mathbf{T}$ together with the source covariance $\mathbf{A} = \mathbf{T}^{-1}\boldsymbol{\Sigma}^*\mathbf{T}^{-\top}$ induces the same transmit covariance $\boldsymbol{\Phi}^*$, and hence achieves the same objective value. (Note that if $\tilde{\mathbf{G}}$ has repeated singular values, $\mathbf{V}_r$ is itself non-unique up to orthogonal transformations within the corresponding singular subspaces.) The result above should therefore be read as identifying one representative element of the optimal set.

*Remark* B.4 (Optimal Power Allocation). The proof of Theorem B.1 does not require the explicit values of the diagonal entries of $\mathbf{\Gamma}^*$. For completeness, they are given by the standard waterfilling solution (Cover & Thomas, 2005):

$$\max_{\mathbf{\Gamma}_{ii} \geq 0, \sum_i \mathbf{\Gamma}_{ii} = P} \prod_i \left( \sigma_i^2 \mathbf{\Gamma}_{ii} + 1 \right),$$

which allocates more power to directions with larger singular values $\sigma_i$ of the whitened channel matrix $\tilde{G}$.

## B.4. Choice of Subspace Basis

While the right singular vectors of $\tilde{G}$ provide a principled and theoretically optimal characterization of the synergy subspace, they are not always the most convenient basis for use in learning algorithms. In particular, singular vectors are defined only up to sign, and their ordering depends on the corresponding singular values. As a result, small changes in $\tilde{G}$—for example due to variation in the state $s_t$ or updates to the dynamics model parameters $\psi$—can induce discontinuous changes in the basis, such as sign flips or reordering of vectors. Moreover, repeated SVD computation during training can be computationally expensive for large matrices.

To address these practical considerations, we adopt a more stable and well-behaved basis for the same subspace. Specifically, we use the rows of $\tilde{G}$, normalized to have unit norm. These vectors span the same subspace as the right singular vectors $V_r$, while varying continuously with $\tilde{G}$ and avoiding the ambiguities inherent to the SVD representation. By Remark B.3, any basis spanning the same subspace as $V_r$ is empowerment-optimal; the row-normalised basis therefore inherits this optimality while varying continuously with $\tilde{G}$.

## C. Musculoskeletal Dynamics

The dynamics of the musculoskeletal system are typically modeled as a third-order system combining first-order muscle activation and contractile dynamics with second-order body dynamics (Caggiano et al., 2022b):

$$\dot{a} = (u - a)/\tau(u, a)$$

$$F_m = a \cdot f_a(l) \cdot f_v(\dot{l}) + f_p(l)$$

$$F_{mtu} = F_m \cos \alpha$$

$$M(q) \cdot \ddot{q} + C(q, \dot{q}) \cdot \dot{q} + g(q) = J^\intercal F_{mtu},$$

where $a$ represents the muscle activation and $\dot{a}$ its time derivative; $u$ represents the neural control signal (the action in our CMP terminology); $\tau(u, a)$ is the time constant of the muscle activation and deactivation dynamics; $F_m$ represents the muscle force; $f_a$ is the active muscle force-length function; $l$ represents the muscle length and $\dot{l}$ its time

derivative; $f_v$ is the muscle force-velocity function; $f_p$ is the passive muscle force-length function; $F_{mtu}$ represents the muscle-tendon unit force; $\alpha$ represents the pennation angle between tendon and muscle fibers; $M$ is the mass/inertia matrix of the skeleton system; $q, \dot{q}, \ddot{q}$ are the joint angles, velocities, and accelerations; $C(q, \dot{q})$ is the Coriolis force term; $g(q)$ is the gravity force term; and the Jacobian transpose $J^\intercal$ maps force vectors from muscle to generalized joint coordinates.

The joint accelerations are an affine function of the activations. Because the MuJoCo simulator updates joint velocities via (semi-implicit) Euler integration ($\dot{q}_{t+1} = \dot{q}_t + \ddot{q}_t \Delta t$), and integration is a linear operation, the discrete-time joint velocities are correspondingly affine in the activations. Consequently, if $\tau(u, a)$ is independent of the neural control signals (i.e., $\tau(u, a) \equiv \tau(a)$), the next-step joint velocities are also an affine function of the controls, making the control-affine modelling assumption in Equation (1) exact. Often, a different time constant is used for muscle activation versus deactivation:

$$\tau(u, a) = \begin{cases} \tau_{\text{act}}(a) & u - a > 0 \\ \tau_{\text{deact}}(a) & u - a \leq 0, \end{cases}$$

in which case the dynamics are piecewise affine in the control signal, and our control-affine assumption remains a robust approximation for the effects of neural control signals on the joint velocities.

## D. Experiments

### D.1. Environments

With the exception of BaodingBalls and DieReorient, we use the default MyoSuite environment settings for all tasks, including reward function parameters and episode lengths. BaodingBalls and DieReorient were the two tasks featured in the NeurIPS MyoChallenge 2022 competition (Caggiano et al., 2022a), where participants were required to design custom reward functions. For these tasks, we adopt the environment configurations and reward parameters used in the Lattice paper (Chiappa et al., 2023). The authors of that work were members of the team that achieved first place in the BaodingBalls challenge, making these settings a strong and well-motivated reference for evaluation.

**Sparse Rewards.** In addition to these standard environments, we conduct a targeted ablation to examine the effect of reward sparsity on manifold discovery and task learning. For this experiment, we modify the dense reward function of Reorient8 to a sparse success signal: the agent receives a reward of 1.0 in solved states and 0.0 otherwise. We retain the environment's default success definition, under which a state is considered solved when the cosine similar-

ity between the current and goal object orientation vectors exceeds a fixed threshold of 0.95.

## D.2. Play-Based Manifold Discovery Protocol

For the experiments in Section 10.3, we follow the protocol introduced in (Berg et al., 2023). A SAC agent is first trained for 1M environment steps on the Reorient8 task, producing a play-phase buffer consisting of state–action–next-state transitions. SAR and JoSEPi use the same play phase to acquire an action manifold, but differ in how it is identified and later exploited during downstream learning.

### D.2.1. SAR

In SAR, muscle activation time series generated by the play-phase policy are collected and analyzed offline. Principal component analysis (PCA) and independent component analysis (ICA) are jointly applied to the time series data (ICAPCA), a technique commonly used in motor neuroscience to identify muscle synergies. The first 20 synergies, capturing over 80% of the variance, are used to define a low-dimensional action subspace.

The SAR policy outputs two action vectors: a low-dimensional synergy-based action and a directly parameterized high-dimensional action. These are combined via a convex combination, with weights 0.66 for the synergy-based action and 0.34 for the direct action. The code used to generate the synergies and policy of SAR was taken from the MyoSuite GitHub repository.

### D.2.2. STATE-DEPENDENT SAR (SD-SAR)

We also consider a variant of SAR (SD-SAR) that uses a state-dependent generalization of PCA to derive a synergy subspace. To motivate our approach, we first note that PCA can be framed as a representation learning problem defined over a data distribution. Specifically, consider a linear autoencoder with encoder matrix $E \in \mathbb{R}^{k \times d}$ and decoder matrix $D \in \mathbb{R}^{d \times k}$ constrained to be orthonormal ($D^\top D = I$). For a zero-mean data distribution $p(x)$, training this autoencoder to minimize the expected Euclidean reconstruction error $\langle \|x - DEx\|^2 \rangle_{p(x)}$ recovers an orthonormal basis for the subspace spanned by the top $k$ principal components of the distribution, given by the columns of the optimal decoder $D$ (Bourlard & Kamp, 1988; Baldi & Hornik, 1989). Moreover, for any fixed orthonormal decoder, the optimal encoder matrix is given in closed form by $E = D^\top$. Substituting this optimal encoder back into the expected loss simplifies the system to the standard population PCA projection objective:

$$\min_{D} \left\langle \|x - DD^\top x\|^2 \right\rangle_{p(x)} \quad \text{s.t.} \quad D^\top D = I.$$

We generalize this population framework to a state-

dependent setting by replacing the state-independent decoder $D$ with a state-conditioned decoder $D(s) \in \mathbb{R}^{d \times k}$. We parameterize the mapping $\xi : \mathcal{S} \to \mathbb{R}^{d \times k}$ from the state $s$ to the decoder matrix via a nonlinear neural network. Specifically, the network outputs a raw matrix $\tilde{D} \in \mathbb{R}^{d \times k}$, and the orthonormal decoder $D(s)$ is constructed by orthonormalizing the columns of $\tilde{D}$ using the Modified Gram–Schmidt procedure.

Because $D(s)$ is orthonormal by construction for any state, the optimal state-dependent encoder matrix remains available in closed form as $E(s) = D(s)^\top$. To train the network using standard stochastic optimization, we minimize the generalized reconstruction objective defined as an expectation over pairs of states and data vectors sampled from the joint distribution:

$$\mathcal{L}(\xi) = \left\langle \|x - D(s)D(s)^\top x\|^2 \right\rangle_{p(s,x)}.$$

We train the decoder network using offline data consisting of concurrent states and muscle activations (the target data $x$) collected during a dedicated play phase (Section 10.3). We train the model for 50K iterations (enough for convergence), where each iteration consists of 20 gradient steps on batches of 256 state-activation pairs sampled uniformly from the play-phase buffer. Network hyperparameters are listed in Table 1.

*Table 1.* **SD-SAR decoder network hyperparameters.**

| parameter | |
|---|---|
| learning rate | $3.0 \times 10^{-4}$ |
| hidden units | [400, 300] |

### D.2.3. JoSEPI

In JoSEPi, the play-phase experience is used to train a dynamics model that serves as the basis for action manifold discovery. The dynamics network is trained on transitions sampled from the play-phase replay buffer (see Appendix E.1 for details). The dynamics model is then held fixed, inducing a static action manifold that is used to learn the Reorient100 task downstream.

## E. Implementation

For Lattice-SAC and Lattice-rPPO, we use the authors' publicly available implementation on GitHub. All other methods (JoSEPi, SAR, SAC) are based on the Stable-Baselines3 JAX (SBX) (Raffin et al., 2021) implementation of SAC (hyperparameters in Table 2). We parallelize agent-environment interactions across 20 CPU cores to accelerate training.

## E.1. Dynamics Model

We consider two variants of the control-affine dynamics model in Equation (1): a full-rank parameterization of $\boldsymbol{G}$ and a low-rank approximation. Let $n = \dim(\mathcal{Y})$ and $m = \dim(\mathcal{A})$ denote the number of joints and muscles, respectively.

### E.1.1. LOW-RANK DYNAMICS APPROXIMATION

Due to biomechanical coupling between joints—arising from shared musculature and tendon networks—muscle commands cannot independently control all joint velocities. Consequently, the input matrix $\boldsymbol{G}$ is generally not full rank.

In the low-rank variant, we parameterize the input matrix as a product of factors:

$$\boldsymbol{G} \approx \boldsymbol{LR},$$

where $\boldsymbol{L} \in \mathbb{R}^{n \times k}$ and $\boldsymbol{R} \in \mathbb{R}^{k \times m}$, with rank $k < n < m$. The rank $k$ roughly corresponds to the intrinsic dimensionality of independent actuation directions, or muscle synergies.

Unless stated otherwise, we use the low-rank model with $k = 15$, which yields slightly improved performance compared to the full-rank parameterization. In Section G.2, we present the results of a sensitivity analysis in which we sweep over $k$.

### E.1.2. DYNAMICS NETWORK

We parameterize the model using a neural network $\psi$ that maps the state $\boldsymbol{s}_t$ to the dynamics parameters. In both cases, the network outputs $\boldsymbol{f} \in \mathbb{R}^n$ and $\tilde{\boldsymbol{q}} \in \mathbb{R}^n$. The covariance matrix is modeled as diagonal and constructed as $\boldsymbol{Q} = \mathrm{diag}(\mathrm{softplus}(\tilde{\boldsymbol{q}})^2)$. The parameterization of the input matrix $\boldsymbol{G} \in \mathbb{R}^{n \times m}$ depends on the variant:

- **Full-rank:** The network outputs $\boldsymbol{G} \in \mathbb{R}^{n \times m}$ directly. The mapping is $\psi : \mathcal{S} \to \mathbb{R}^n \times \mathbb{R}^{n \times m} \times \mathbb{R}^n$.

- **Low-rank:** The network outputs the factors $\boldsymbol{L}$ and $\boldsymbol{R}$. The mapping is $\psi : \mathcal{S} \to \mathbb{R}^n \times \mathbb{R}^{n \times k} \times \mathbb{R}^{k \times m} \times \mathbb{R}^n$. The input matrix is then reconstructed as $\boldsymbol{G} = \boldsymbol{LR}$.

Training details are described below.

### E.1.3. TRAINING PROCEDURE

Training is performed via maximum likelihood on a dataset of transitions $\mathcal{D} = \{(\boldsymbol{s}^{(i)}, \boldsymbol{a}^{(i)}, \boldsymbol{y}'^{(i)})\}_{i=1}^n$ (e.g., a replay buffer), where each transition consists of a state $\boldsymbol{s}$, an action $\boldsymbol{a}$, and the next joint velocities $\boldsymbol{y}' = h(\boldsymbol{s}')$. Optimization is carried out using the ADAM optimizer (Kingma & Ba, 2015), with transitions sampled uniformly with replacement. We use MC dropout (Gal & Ghahramani, 2016) as

a lightweight approximation to Bayesian inference in deep networks. Dropout induces an implicit variational posterior $q_\psi(\phi | \boldsymbol{s}_t, \mathcal{D})$ over the state-dependent dynamics parameters $\phi \triangleq \{\boldsymbol{f}, \boldsymbol{G}, \tilde{\boldsymbol{q}}\}$. Sampling different dropout masks provides a computationally efficient estimate of epistemic uncertainty, inducing a corresponding distribution over the optimal precoder (Equation (2)). Network hyperparameters are listed in Table 3.

### E.1.4. SIMULTANEOUS MANIFOLD AND TASK LEARNING

When simultaneously learning the dynamics and SAC networks (policy and Q-functions) (Sections 10.1 and 10.2), all networks are trained using the same shared replay buffer. The dynamics network is trained with the same training frequency, number of gradient steps, and batch size as the SAC networks (Table 2). At each training iteration, we first update the dynamics network and then update the SAC networks, ensuring that the policy and Q-functions are optimized with respect to the most up-to-date action manifold.

### E.1.5. LEARNING DYNAMICS FROM PLAY

In the play setting (Section 10.3), we train the dynamics network using offline data collected during a dedicated play phase. We train the model for 1M iterations, where each iteration consists of 20 gradient steps on batches of 256 transitions sampled uniformly from the play-phase buffer.

*Table 2.* **SAC hyperparameters.** The same parameters were used in all experiments and by all models.

| parameter | |
|---|---|
| policy learning rate | $3.0 \times 10^{-4}$ |
| Q-function learning rate | $3.0 \times 10^{-4}$ |
| policy hidden units | [400, 300] |
| Q-function hidden units | [400, 300] |
| buffer size | 300,000 |
| batch size | 256 |
| soft update coefficient | 0.02 |
| discount factor | 0.98 |
| learning starts (steps) | 3,000 |
| train frequency (steps) | 1 |
| gradient steps | 20 |
| target entropy | $-\dim(\mathrm{support}(\pi))$ |
| normalize obs | true |

# F. Empirical Evaluation of the Dynamics Model

To rigorously evaluate both the accuracy of our control-affine dynamics approximation and the calibration of our epistemic uncertainty estimates, we conduct a post-hoc sta-

*Table 3.* **Dynamics network hyperparameters.** *For simultaneous manifold and task learning, the learning rate was linearly decayed to 0 over the course of training, whereas when training from the static play dataset a constant learning rate was used.

| parameter | value |
|---|---|
| learning rate* | $1.0 \times 10^{-5}$ |
| hidden units | $[400, 300]$ |
| dropout rate | $[0.5, 0.5]$ |

tistical analysis of the dynamics model predictions. The dataset used for this analysis is constructed by executing the trained policy from each environment and training run in Section 10.1 for 100 episodes and aggregating all transitions. At each transition $(\boldsymbol{s}, \boldsymbol{a}, \boldsymbol{y}')$, we draw $K = 50$ samples $\hat{\phi}^{(k)} \sim q_\psi(\phi|\boldsymbol{s}, \mathcal{D})$ from the implicit variational posterior of the dynamics parameters. These samples were used to construct a predictive distribution over the next joint velocities:

$$p(\boldsymbol{y}'|\boldsymbol{s}, \boldsymbol{a}) = \frac{1}{K} \sum_{k=1}^{K} p(\boldsymbol{y}'|\boldsymbol{s}, \boldsymbol{a}, \hat{\phi}^{(k)}).$$

For the $i$-th transition in the dataset, $(\boldsymbol{s}^{(i)}, \boldsymbol{a}^{(i)}, \boldsymbol{y}'^{(i)})$, we defined:

1. the **prediction error** ($e^{(i)}$) as the mean squared error (MSE) between the expected next joint velocities $\hat{\boldsymbol{y}}'^{(i)} = \langle \boldsymbol{y}' \rangle_{p(\boldsymbol{y}'|\boldsymbol{s}^{(i)}, \boldsymbol{a}^{(i)})}$ and the true next joint velocities $\boldsymbol{y}'^{(i)}$, averaged across the $D$ joint velocity dimensions:

$$e^{(i)} = \frac{1}{D} \left\| \hat{\boldsymbol{y}}'^{(i)} - \boldsymbol{y}'^{(i)} \right\|_2^2;$$

2. the **predictive variance** ($v^{(i)}$) as the trace of the predictive covariance matrix divided by $D$:

$$v^{(i)} = \frac{1}{D} \sum_{d=1}^{D} \text{Var}_{p(\boldsymbol{y}'|\boldsymbol{s}^{(i)}, \boldsymbol{a}^{(i)})} [\boldsymbol{y}'_d].$$

This represents the average epistemic uncertainty of the model across the joint velocity dimensions;

3. the **contact switch label** ($c^{(i)}$) as a binary indicator where $c^{(i)} = 1$ if at least one fingertip switches between contact and non-contact states during the transition, and $c^{(i)} = 0$ otherwise.

Computing these metrics across all transitions yielded a final evaluation pool of $N = 273,940$ tuples, $(e^{(i)}, v^{(i)}, c^{(i)})$, which serves as the foundation for the following analyses.

## F.1. Accuracy of the Control-Affine Approximation

Contact-rich manipulation tasks are characterized by intermittent contact events that introduce local discontinuities into the system dynamics. To assess how well the control-affine approximation in Equation (1) handles these events, we stratify the evaluation pool based on the contact switch label $c$.

As shown in Table 4, the mean prediction error (mean $\pm$ s.e.m.) is $e = 0.172 \pm 0.001$ at transitions with a contact switch compared to $0.150 \pm 0.001$ without—a small difference in means relative to the within-group variability (s.t.d. $= 0.466$ and $0.334$, respectively). A linear mixed-effects model with a fixed effect of the contact switch label and random intercepts for each experimental run (environment–seed combination) confirms that while the large sample size ($N = 273,940$) yields a highly significant effect ($\beta = 0.056$, s.e. $= 0.002$, $z = 31.38$, $p < 1 \times 10^{-16}$), the increase in prediction error is modest. Taken together, these results suggest that the model predictions remain largely stable across task phases, with limited degradation during contact switching events, verifying that the control-affine assumption serves as a reliable basis for optimizing JoSE. We hypothesize that the contact discontinuities are gracefully accommodated by the control-affine formulation as it allows for arbitrary nonlinearities with respect to the state. We also speculate that by incorporating richer contact-related observations, such as tactile or force feedback, prediction errors during contact switches could be reduced further.

*Table 4.* **Dynamics model prediction error grouped by contact switch label.** Data pooled across all environments, training runs, episodes, and time steps ($N = 273,940$), stratified by whether a contact switch occurred ($c = 1$) or not ($c = 0$).

| switch | mean | s.e.m. | count | s.t.d. |
|---|---|---|---|---|
| yes | 0.172 | 0.001 | 180,115 | 0.466 |
| no | 0.150 | 0.001 | 93,825 | 0.334 |

## F.2. MC Dropout Uncertainty Calibration

Using the same evaluation dataset, we examine whether the epistemic uncertainty represented by the MC dropout ensemble is meaningful in practice. Specifically, we calculate the Pearson correlation coefficient between the predictive variance $v$ and the prediction error $e$ across all tuples.

As shown in Table 5, a consistent positive correlation ($r \in [0.51, 0.65]$) is observed across all environments, indicating that the dropout ensemble provides a meaningful signal about regions of high model uncertainty. This correlation is remarkably high given the lightweight nature of the training procedure (single-mask dropout was used at training time) and the high-dimensional nature of the dynamics network

*Table 5.* **Correlation between MC dropout predictive variance and prediction error.** Data show Pearson correlation coefficient (mean ± s.e.m. across 5 training runs).

| Environment | Pearson $r$ |
|---|---|
| DieReorient | $0.651 \pm 0.015$ |
| BaodingBalls | $0.509 \pm 0.074$ |
| KeyTurn | $0.569 \pm 0.016$ |
| PenTwirl | $0.556 \pm 0.042$ |

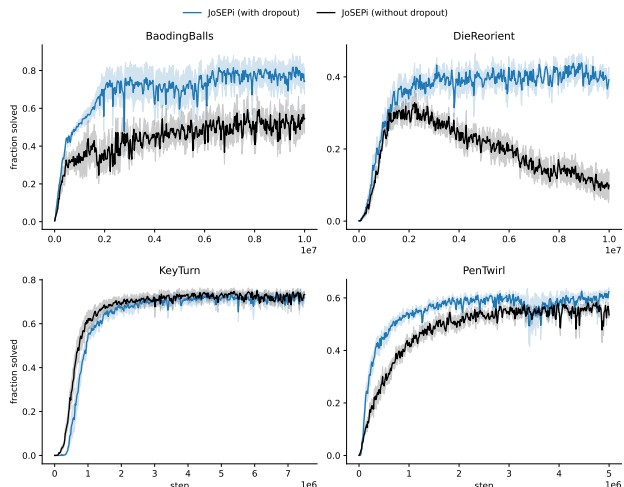

*Figure 7.* **MC dropout ablation:** Performance of JoSEPi with different per-layer MC dropout rates in the dynamics model ($p = 0.5$, blue; $p = 0.0$, black) on BaodingBalls, DieReorient, KeyTurn, and PenTwirl. Data show solved fraction over training (mean ± s.e.m. across 5 random seeds).

input (the state spaces of the MyoHand and each rigid-body object are 85- and 13-dimensional, respectively).

It is important to note that the predictive variance specifically isolates epistemic uncertainty rather than total prediction error. Systematic errors stemming from model mismatch are irreducible and do not inflate MC dropout variance: under persistent model bias, all dropout samples converge to the same biased prediction, exhibiting zero epistemic variance. The correlation observed therefore reflects reducible uncertainty from limited data—precisely the type of metric required to drive intrinsically motivated exploration.

# G. Ablations and Sensitivity Analyses

## G.1. Effect of Dynamics Uncertainty

To isolate the impact of representing dynamics uncertainty, we set the MC dropout rate to $0.0$ (thereby learning a single dynamics model rather than an ensemble) and repeat the benchmark evaluation from Section 10.1. We find that modeling epistemic uncertainty via MC dropout substantially enhances both asymptotic performance and training stability (Figure 7).

## G.2. Effect of Input Matrix Factorization Rank

The factorization rank $k$ of the input matrix $\boldsymbol{G}$ (Section E.1.1) directly determines the dimensionality of both the latent space and the learned action manifold. To characterize the sensitivity of JoSEPi to the manifold dimensionality, we sweep over $k \in \{9, 12, 15, 18, 23\}$ and repeat the benchmark evaluation from Section 10.1. As shown in Table 6, performance remains remarkably stable across a wide range of dimensionalities ($k \geq 12$). The first meaningful performance drop occurs at $k = 9$; this degradation is most pronounced on BaodingBalls, a particularly challenging task that requires in-hand manipulation of multiple objects. Although we currently treat $k$ as a user-specified hyperparameter, future extensions could automate this selection, for example, by inspecting the singular values of the full-rank dynamics matrix $\tilde{\boldsymbol{G}}$ to determine its effective rank.

*Table 6.* **Effect of input matrix factorization rank on performance.** Data show solved fraction at 5M steps of training (mean ± s.e.m. across 5 random seeds) as a function of input matrix rank $k$.

| $k$ | BaodingBalls | DieReorient | KeyTurn | PenTwirl |
|---|---|---|---|---|
| 23 | $0.572 \pm 0.052$ | $0.355 \pm 0.023$ | $0.688 \pm 0.023$ | $0.609 \pm 0.016$ |
| 18 | $0.687 \pm 0.067$ | $0.344 \pm 0.020$ | $0.750 \pm 0.013$ | $0.600 \pm 0.011$ |
| 15 | $0.611 \pm 0.102$ | $0.380 \pm 0.031$ | $0.717 \pm 0.021$ | $0.624 \pm 0.015$ |
| 12 | $0.700 \pm 0.056$ | $0.357 \pm 0.015$ | $0.711 \pm 0.014$ | $0.619 \pm 0.009$ |
| 9 | $0.378 \pm 0.075$ | $0.284 \pm 0.030$ | $0.641 \pm 0.029$ | $0.556 \pm 0.012$ |

# H. Muscle Synergy Analysis via NMF

To connect our findings to the motor control literature on muscle synergies, we apply standard electromyography (EMG) decomposition techniques to the muscle activations generated by policies constrained to JoSE-optimized action manifolds. Specifically, for each benchmark environment in Section 10.1, we execute the policy at 8 equally spaced intervals throughout training and apply non-negative matrix factorization (NMF) to the resulting muscle activations, sweeping over the number of synergies $k \in \{3, 6, 9, 12, 15, 18\}$. The `scikit-learn` implementation of NMF is used. For each value of $k$, we evaluate the variance accounted for (VAF), defined as:

$$\text{VAF} = \left( 1 - \frac{\sum_{i,j}(\boldsymbol{X}_{ij} - \hat{\boldsymbol{X}}_{ij})^2}{\sum_{i,j}(\boldsymbol{X}_{ij} - \bar{\boldsymbol{X}}_j)^2} \right) \times 100\%, \quad (3)$$

where $\boldsymbol{X} \in \mathbb{R}^{N \times M}$ represents the empirical muscle activation matrix for $N$ samples and $M$ muscles, $\hat{\boldsymbol{X}} = \boldsymbol{W}\boldsymbol{H}$

is the low-rank NMF reconstruction given coefficients $W \in \mathbb{R}^{N \times k}$ and synergies $H \in \mathbb{R}^{k \times M}$, and $\bar{X}_j$ is the mean activation of muscle $j$ across all samples. Identifying the point of VAF saturation is the standard approach used to infer the number of synergies in the literature.

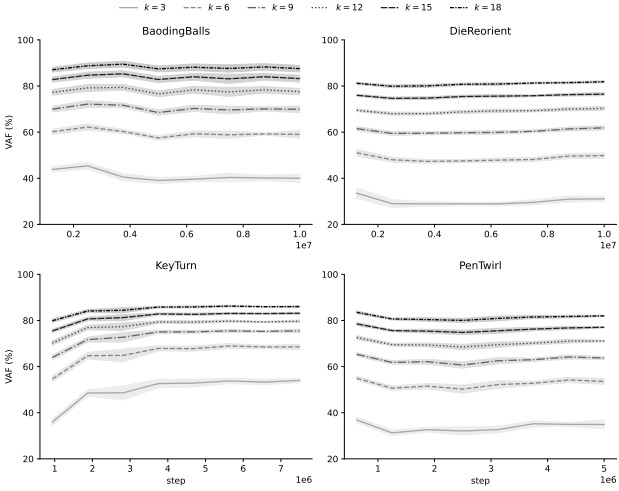

*Figure 8.* **Muscle synergy decomposition over training.** Data show variance accounted for (VAF) (mean $\pm$ s.e.m. across 5 training runs) of muscle activations by non-negative matrix factorization (NMF) as a function of training step for Baoding-Balls, DieReorient, KeyTurn, and PenTwirl. Lines within each panel track a hyperparameter sweep over the number of synergies $k \in \{3, 6, 9, 12, 15, 18\}$.

To correctly interpret the NMF results, it is important to consider how muscle activations are generated. At each time step, the policy outputs a muscle command that is confined to a *state-dependent subspace*. This command drives a change in the muscle activations, which in turn generate forces that alter the state of the system. Consequently, at the next time step, the policy outputs a command within a reoriented state-dependent subspace. Because the system traverses many states over the course of a trajectory, the resulting muscle commands, and their downstream activations, can span a much higher dimensional space than each state-specific subspace allows. Indeed, despite JoSEPi outputting commands within a 15-dimensional subspace in every state, NMF analysis finds that 15 synergies account for only 76–84% of the total variance of the muscle activations (Figure 8). This has implications for biological motor control: the central nervous system may use fewer synergies than traditional state-blind analyses suggest, with the apparent high-dimensional complexity of behavior emerging from state-dependent reorientation of low-dimensional control subspaces.

Viewed through this lens, the evolution of the VAF over

training provides insights into how the state-dependence of the subspace develops. Specifically, a higher VAF suggests a more consistent orientation of the subspace across the state space, whereas a lower VAF implies more extensive state-dependent reorientation. Empirically, the VAF remains exceptionally stable throughout learning, shifting by only 5–10% between the first and second evaluation intervals and remaining largely constant thereafter (Figure 8). This indicates that this state-dependent structure is established early in training, potentially underpinning the rapid concurrent improvements in task performance (Figure 6).

