# OpenReview forum: "Joint-Space Empowerment as a Theory of Dexterous Motor Coordination"
_ICML.cc/2026/Conference — ICML 2026 spotlight_

### Official Review · Reviewer_m61h · 2026-03-03

**Soundness:** 3
**Presentation:** 3
**Significance:** 3
**Originality:** 3
**Overall Recommendation:** 5
**Confidence:** 4

**Summary:**

This paper introduces a framework for learning low-dimensional, state-dependent action manifolds in high-dimensional, overactuated control systems. The authors propose Joint-Space Empowerment (JSE), an information-theoretic objective that maximizes mutual information between latent actions and task-agnostic joint velocities. Under Gaussian dynamic model, they derive a close-form solution for the optimal precoder defining the action manifold. Experiments on dexterous manipulation tasks show strong performance, generalization, and robustness to sparse rewards compared to existing baselines.

**Compliance With Llm Reviewing Policy:**

Affirmed.

**Final Justification:**

I appreciate the authors successfully resolving my concerns in the rebuttal, and I will maintain my score of 5.

**Key Questions For Authors:**

1. The introduction provides strong biological motivation, but does not clearly present the specific machine learning challenge or technical novelty. For example, the transition to Joint-Space Empowerment (JSE) feels abrupt, with limited intuition about its necessity and distinction from prior empowerment or representation-learning methods. Clarifying the RL difficulty due to high-dimensional action redundancy and introducing the core theoretical and algorithmic contributions would strengthen the presentation.

2. Have you tested MANIP on other overactuated systems beyond MyoHand? To what extent do you expect the approach to generalize to non-musculoskeletal or non-biomechanical control domains?

3. How sensitive is performance to the choice of intrinsic dimension k? Although k is described as reflecting the intrinsic actuation dimensionality, it appears to be chosen empirically (k=15). Is there an ablation study over different values of k, and could this dimension be selected or adapted automatically in other systems?

**Limitations:**

The paper does not raise obvious negative societal impact concerns. However, the discussion of technical limitations could be strengthened. In particular, the authors could more explicitly discuss the reliance on the affine-Gaussian dynamics assumption, potential scalability issues, and the extent to which the approach generalizes beyond the MyoHand setting.

**Strengths And Weaknesses:**

Strengths:
1. The theoretical formulation is interesting and principled: it frames manifold learning as an information-theoretic optimal precoding problem, enabling the discovery of structured action manifolds from a learned dynamics model and integrating them naturally into a reinforcement learning framework.
2. The experimental evaluation is convincing, with strong empirical performance across manipulation tasks.

Weaknesses:
1. The introduction does not fully articulate the machine learning motivation and technical novelty, and could more clearly position the work relative to prior RL and empowerment-based approaches.

---

> ### Author Rebuttal · Authors · 2026-03-31
>
> We thank the Reviewer for their constructive feedback and for recognizing the "interesting and principled" nature of our theoretical formulation. We are encouraged by your assessment of our experimental evaluation as "convincing".
>
> > Contribution to Machine Learning
>
> Following your suggestions, we have refined the manuscript to better articulate the machine learning motivations and technical novelty.
>
> Our study focuses on solving the fundamental problem of action-redundancy [1-5] — a primary bottleneck for reinforcement learning in large discrete action spaces, high-dimensional continuous action spaces, and overactuated robots. Action redundancy complicates both value-based and policy-gradient reinforcement learning methods, reducing the sample-efficiency of learning. For value-based methods, treating behaviorally equivalent actions as distinct dilutes the learning signal across the action space [1-4]. For policy-gradient methods, action redundancy increases the variance of gradient estimates [5]. Previous approaches have addressed this problem by utilizing action-elimination architectures to prune redundant candidates from the policy's search space [1], mapping actions into latent embeddings to group behaviorally similar outcomes [2,3], or employing action-dependent baselines to isolate the marginal contribution of redundant factors [5]. In contrast, we address action redundancy by utilizing an information-theoretic objective for discovering a state-dependent action subspace that provides maximum influence over a controlled variable of interest. Moreover, we develop an efficient, model-based, uncertainty-aware algorithm for discovery the manifold.
>
> > Beyond the MyoHand
>
> We have previously applied our framework to a musculoskeletal model of the human upper limb, featuring 27 Degrees of Freedom and 63 muscles. MANIP successfully learned to reach for, grasp, and relocate objects. Our approach achieved 1st place in a recent major international machine learning competition focused on biomechanical control (specific details are omitted to preserve anonymity but will be included in the final manuscript). This gives us confidence that MANIP is highly scalable, and we are currently in the process of applying it to full body musculoskeletal models with hundreds of muscles.
>
> As our method is derived from general information-theoretic principles, we expect it to generalize to any overactuated system—such as cable-driven robots or soft actuators. While the control-affine approximation’s utility in non-biomechanical domains remains an empirical question, we provide 2 distinct contributions: (1) a novel information-theoretic objective for discovering task-agnostic control subspaces, and (2) a practical algorithm for integrating these subspaces into RL. We hope our objective inspires future researchers to develop alternative manifold-learning algorithms tailored to diverse systems.
>
> > Dimensionality Ablation
>
> We performed a sweep over the rank $k \in$ {9, 12, 15, 18, 23} hyperparameter and found that performance remains remarkably stable from $k$ = 12 to $k$ = 23 (Table 1). The first significant drop in performance is seen at $k$ = 9 (most notable in the two-object BaodingBalls task).
>
> #### Table 1: Solved Fraction at 5M Steps (Mean ± SEM)
>
> | Rank | BaodingBalls | DieReorient | KeyTurn | PenTwirl |
> |---|---|---|---|---|
> | k = 23 | 0.572 ± 0.052 | 0.355 ± 0.023 | 0.688 ± 0.023 | 0.609 ± 0.016 |
> | k = 18 | 0.687 ± 0.067 | 0.344 ± 0.020 | 0.750 ± 0.013 | 0.600 ± 0.011 |
> | k = 15 | 0.611 ± 0.102 | 0.380 ± 0.031 | 0.717 ± 0.021 | 0.624 ± 0.015 |
> | k = 12 | 0.700 ± 0.056 | 0.357 ± 0.015 | 0.711 ± 0.014 | 0.619 ± 0.009 |
> | k = 9 | 0.378 ± 0.075 | 0.284 ± 0.030 | 0.641 ± 0.029 | 0.556 ± 0.012 |
>
> The selection of $k$ could be automated by analyzing the singular values of the full-rank dynamics matrix and determining its effective rank. This approach is well suited to Learning-To-Learning, where $k$ can be set prior to downstream learning.
>
> > Neuroscience Contributions
>
> Although MANIP operates within a 15D state-dependent subspace, traditional Non-negative Matrix Factorization (NMF) of the muscle activations found that 15 synergies only account for 76–84% of the variance. This is because the union of the state-dependent subspaces spans a higher-dimensional volume when marginalized across the state space. This suggests that the brain may utilize far fewer fundamental synergies than previous state-independent analyses have assumed.
>
> #### References
>
>  [1]. Zahavy, T. et al. (2018). Learn What Not to Learn: Action Elimination with Deep Reinforcement Learning. NeurIPS.
>
>  [2]. Dulac-Arnold et al. (2015). Deep Reinforcement Learning in Large Discrete Action Spaces. arXiv.
>
>  [3]. Tennenholtz et al. (2019). The Natural Language of Actions. ICML.
>
>  [4]. Baram et al. (2021). Action Redundancy in Reinforcement Learning. ICML.
>
>  [5]. Wu, C. et al. (2018). Variance Reduction for Policy Gradient with Action-Dependent Factorized Baselines. ICLR.

---

> > ### Author Rebuttal · Reviewer_m61h · 2026-04-01
> >
> > The authors have successfully addressed my previous concern regarding the introduction. The revised text now clearly articulates the machine learning motivation and technical novelty. I believe this paper is a valuable contribution to the community, and I recommend for acceptance.

---

> > > ### Author Response · Authors · 2026-04-02
> > >
> > > Thank you for upgrading your rating. Your questions, comments and insights have been invaluable. We will incorporate the new text and experimental results that you requested, and we believe this will significantly strengthen the paper.

---

### Official Review · Reviewer_TRTY · 2026-03-11

**Soundness:** 3
**Presentation:** 3
**Significance:** 3
**Originality:** 4
**Overall Recommendation:** 5
**Confidence:** 4

**Summary:**

The authors propose a framework for motor coordination in musculoskeletal systems using the information-theoretic principle of empowerment. The core idea is to maximize mutual information between an agent actions and its future joint-velocity states, identifying a low-dimensional action manifold that simplifies reinforcement learning in overactuated systems. To formalize this, the authors introduce Joint-Space Empowerment (JSE), a metric quantifying intrinsic control over a physical embodiment. By extending JSE to Controlled Markov Processes, they provide a mathematical basis for applying empowerment-based objectives to a broad class of motor control problems.

The key technical contribution is the Manifold-Precoder algorithm (MANIP), which bridges JSE theory and practical experimentation. MANIP learns a control-affine dynamics model mapping muscle commands to joint angular velocities and extracts the empowerment-maximizing action manifold in closed form via the SVD of the whitened control matrix. The resulting manifold can be learned simultaneously with task policy training or pre-learned during a task-agnostic play phase to accelerate downstream adaptation. The approach is validated on high-dimensional biomechanical manipulation tasks using musculoskeletal hand models from MyoSuite, where MANIP consistently outperforms chosen baselines in sample efficiency, asymptotic performance, and robustness to reward sparsity.

**Compliance With Llm Reviewing Policy:**

Affirmed.

**Final Justification:**

After reading the author rebuttal and the responses to all reviewers, I maintain my recommendation of 5: Accept and my confidence of 4.

**Key Questions For Authors:**

1. The dynamics model in Eq. (1) linearizes the muscle-command-to-joint-velocity relationship. In contact-rich tasks like BaodingBalls and DieReorient, impulsive forces during object transitions are precisely where this approximation is weakest. Could the authors provide one-step prediction error of the learned model broken down by task phase, and in particular during contact events? A response quantifying the approximation quality in these phases would significantly affect the evaluation of soundness.

2. How does performance degrade as the dimension $k$ of the action manifold is reduced below the reported value of 15? Is there an identifiable threshold below which control becomes infeasible for this system? A systematic ablation over $k$ would clarify whether the closed-form rank choice is robust or fragile.

3. Monte Carlo dropout is used as a posterior approximation over dynamics parameters, inducing a distribution over optimal precoders. Could the authors report whether the uncertainty estimated by the dropout model is meaningful in practice, for example by checking whether higher variance across dropout samples correlates with states where the dynamics model fits poorly? It was not clear from the paper how a poorly calibrated posterior would affect the quality of the sampled precoders.

4. In Figure 3, the advantage of MANIP over baselines appears to diminish at longer training horizons for this specific task, unlike the other three tasks. Do the authors have a hypothesis for why this occurs? Understanding this case would help characterize when JSE is most and least beneficial.

**Limitations:**

The paper would benefit from a dedicated section discussing the limitations of the proposed method. Specifically, the authors should address the sensitivity of their approach to the accuracy of the learned predictive model, the potential impact of reducing the action manifold dimension too aggressively, and any assumptions made in the theoretical derivations (such as the affine mapping between muscle activations and joint velocities). Including empirical or theoretical analysis of these aspects would provide a more balanced perspective and help readers understand the boundaries of the method’s applicability.

**Strengths And Weaknesses:**

**Soundness**

The choice of benchmarks is appropriate and the results provide convincing evidence of the method efficacy. The ablation in Appendix E is informative: the low-rank approximation matches full-rank performance, and dropout uncertainty modeling proves critical for training stability. The central concern I could not resolve from reading the paper is how well the control-affine approximation in Eq. (1) holds during actual manipulation. BaodingBalls and DieReorient involve intermittent contact forces that are precisely the regime where linear dynamics approximations are weakest. The paper does not report one-step prediction error of the learned model across task phases, and I was not able to assess from the text alone whether the model remains accurate during contact transitions.

I did not find a clear derivation in the main text of why joint angular velocities follow an affine model in muscle commands; the argument is deferred to Appendix B, but even there I could not fully verify the step from muscle activation to joint velocity rather than joint acceleration. The use of Monte Carlo dropout as a posterior approximation over dynamics parameters is pragmatic and well-cited, but it was not clear from the paper how a poorly calibrated posterior would affect the quality of the sampled precoders in practice.

**Presentation**

The narrative is clear and the paper is easy to follow overall. The related work section is thorough and the distinction between state-independent subspace methods and the state-dependent manifold of MANIP is well articulated. Two issues stood out. First, several key justifications are placed entirely in the appendix, making the main text harder to evaluate in isolation. In particular, the justification for the control-affine approximation as a reasonable model of musculoskeletal dynamics appears only in Appendix B, while it is central to the validity of the entire framework. Second, the symbol $\delta$ in lines 180-181 lacks a formal definition. I could not determine from context whether this denotes a Dirac distribution or a specific constant, and this should be clarified explicitly.

In Figure 3, the performance of MANIP on KeyTurn converges toward baseline performance at longer training horizons, which is the one result that does not follow the general pattern of sustained superiority. No explanation is offered for this, and I was not able to infer one from the task description alone. An account of why the JSE-driven advantage diminishes for this specific task would help the reader understand the boundary conditions of the method. Visualizing the evolution of the action manifold or the derived synergies across tasks and training stages would also strengthen the paper, both as a validation that the learned synergies are meaningful and as a way to connect the theoretical framework to observed behavior.

**Significance**

The objective of deriving a low-dimensional action manifold for reinforcement learning in overactuated systems addresses a genuine bottleneck in musculoskeletal control. The learning-to-learn regime and the sparse-reward robustness result both demonstrate practically relevant properties beyond the core benchmark comparisons, and the cited downstream applications in assistive robotics and rehabilitation are credible.

The main limitation on significance is that evaluation is restricted to a single simulation platform and morphology. It is not clear how JSE would behave in other overactuated systems such as cable-driven robots or soft actuators, where the control-affine assumption may be even less accurate. The paper also makes broad claims about implications for biological motor control but provides no biological validation. I was unable to assess whether the synergies derived by MANIP resemble those reported in the motor neuroscience literature, for instance via EMG decomposition in human subjects, and the paper does not attempt this comparison.

**Originality**

The derivation of a closed-form solution for the empowerment-maximizing precoder is the paper most significant technical contribution. Casting synergy discovery as an optimal precoding problem is novel in this context and analytically clean. The state-dependent manifold structure, where the synergy subspace varies with the current state rather than being fixed globally, is a concrete advance over prior methods such as SAR that rely on a single static subspace. While empowerment as an intrinsic motivation signal and low-rank action representations are both established ideas, their combination via the precoding framework yields a closed-form, model-based, and sample-efficient solution that represents a non-trivial synthesis.

---

> ### Author Rebuttal · Authors · 2026-03-31
>
> We thank Reviewer TYTY for the positive assessment of our theoretical contribution and the MANIP algorithm’s originality.
>
> > Control-Affine Accuracy
>
> We hypothesize that the control-affine structure can in principle accommodate state-dependent discontinuities without large model errors, as the transition dynamics can be arbitrarily nonlinear with respect to the state.
>
> To assess how well the control-affine approximation holds during manipulation, we examined one-step prediction errors of the learned dynamics model during contact events. We group mean squared prediction errors based on whether at least one finger tip transitioned from contact to no contact or vice versa (Contact Transition) across consecutive timesteps.
>
> The pooled dataset (N = 273,940) showed that the mean squared error (MSE) for a Contact Transition was 0.172 (SEM = 0.001), compared to 0.150 (SEM = 0.001) for states with no Contact Transition (Table 1). The standard deviations in both conditions (0.334 and 0.466, respectively) are large relative to the difference in means, indicating substantial variability within each group.
>
> #### Table 1: MSE
>
> | Contact Transition   |   Mean |   SEM |   Count |   STD |
> |:---------------------|-------:|------:|--------:|------:|
> | No                   |  0.15  | 0.001 |   93825 | 0.334 |
> | Yes                  |  0.172 | 0.001 |  180115 | 0.466 |
>
>
> We employed a linear mixed-effects model with a fixed effect of contact transition and random intercepts for each experimental run (environment–seed combination). The mixed-effects model revealed a statistically significant effect of contact transitions on mean squared error (β = 0.056, SE = 0.002, z = 31.38, p < 1 × 10⁻¹⁶) (Table 2). This result should be interpreted in the context of the very large sample size (N = 273,940), where even small differences can achieve strong statistical significance.
>
> #### Table 2: Mixed Effects
> |                    |   Coef. |   Std. Err. |      z | P           |   CI (2.5%) |   CI (97.5%) |
> |:-------------------|--------:|-----------:|-------:|:------------|----------------:|-----------------:|
> | Intercept          |   0.142 |      0.036 |  3.996 | 6.5 x 10⁻⁰⁵ |           0.073 |            0.212 |
> | Contact Transition |   0.056 |      0.002 | 31.382 | < 1 x 10⁻¹⁶ |           0.052 |            0.059 |
>
>
> The modest increase in prediction error suggests that performance remains largely stable across task phases, with limited degradation during contact transition events. Given that the transition dynamics are nonlinear with respect to the state observation, we speculate that with richer observations (e.g. haptic or force feedback), the prediction errors during contact could be reduced further.
>
> > Dimensionality Ablation
>
> We performed a sweep over the rank $k \in$ {9, 12, 15, 18, 23} hyperparameter and found that performance remains remarkably stable from $k$ = 12 to $k$ = 23 (Table 3). The first significant drop in performance is seen at $k$ = 9 (most notable in the two-object BaodingBalls task).
>
> #### Table 3: Solved Fraction at 5M Steps (Mean ± SEM)
>
> | Rank | BaodingBalls | DieReorient | KeyTurn | PenTwirl |
> |---|---|---|---|---|
> | k = 23 | 0.572 ± 0.052 | 0.355 ± 0.023 | 0.688 ± 0.023 | 0.609 ± 0.016 |
> | k = 18 | 0.687 ± 0.067 | 0.344 ± 0.020 | 0.750 ± 0.013 | 0.600 ± 0.011 |
> | k = 15 | 0.611 ± 0.102 | 0.380 ± 0.031 | 0.717 ± 0.021 | 0.624 ± 0.015 |
> | k = 12 | 0.700 ± 0.056 | 0.357 ± 0.015 | 0.711 ± 0.014 | 0.619 ± 0.009 |
> | k = 9 | 0.378 ± 0.075 | 0.284 ± 0.030 | 0.641 ± 0.029 | 0.556 ± 0.012 |
>
> > MC Dropout Uncertainty
>
> We computed (i) the variance of the one-step model predictions across dropout samples and (ii) the mean squared error between the mean model prediction across dropout samples and the true next joint velocities. A simple linear regression revealed a consistent positive correlation across all tasks (Pearson $r \in [0.51, 0.65]$) (Table 4). This correlation is surprisingly high given the lightweight nature of the training procedure - a single dropout sample is used at training time - and the high-dimensional nature of the joint state-action space of the MyoHand (139-D).
>
> #### Table 4: Pearson Correlation
>
> | Environment | Mean ± SEM |
> |------------|----------------|
> | DieReorient | 0.651 ± 0.015 |
> | BaodingBalls | 0.509 ± 0.074 |
> | KeyTurn | 0.569 ± 0.016 |
> | PenTwirl | 0.556 ± 0.042 |
>
> > KeyTurn Performance
>
> For the KeyTurn task, 2 of the baselines reach asymptotic performance close to (but not better than) MANIP (solved fraction ~0.65-0.7). We hypothesize that KeyTurn represents a different regime of dexterity. Specifically, KeyTurn is primarily a 'grasp-and-rotate' task. Once the initial grasp is secured (by as little as two digits), the requirement for high-dimensional digit coordination diminishes significantly; the turning of the key is predominantly driven by the wrist (a low-DOF joint). Nevertheless, MANIP is still substantially more sample-efficient, converging around 2-3x more quickly.

---

> > ### Author Rebuttal · Reviewer_TRTY · 2026-04-03
> >
> > I thank the authors for their detailed and thorough rebuttal. My main concerns have been addressed as follows. I maintain the score I indicated in my original review, which I believe is fair given the quality of the work and the satisfactory rebuttal.

---

> > > ### Author Response · Authors · 2026-04-03
> > >
> > > We thank you for reaffirming your support for our paper. We are incredibly grateful for your thoughtful insights, critiques and suggestions for further analyses. We look forward to including the results of these analyses in the final paper, as we believe they will significantly strengthen the manuscript.

---

### Official Review · Reviewer_TjYt · 2026-03-11

**Soundness:** 2
**Presentation:** 2
**Significance:** 3
**Originality:** 3
**Overall Recommendation:** 4
**Confidence:** 4

**Summary:**

Overall, the paper studies the problem of discovering effective control policies for high-dimensional, overactuated musculoskeletal systems. The paper introduces the concept of joint-space empowerment (JSE), defined as an information-theoretic principle for quantifying an agent’s control authority over its mechanical degrees of freedom. The paper further proposes a model-based reinforcement learning algorithm, MANIP, and evaluates it on several contact-rich hand manipulation tasks.

**Compliance With Llm Reviewing Policy:**

Affirmed.

**Final Justification:**

Thank you for the detailed rebuttal. The revised terminology and renaming adequately address my concerns regarding the use of “manifold,” and I appreciate the effort to improve conceptual clarity.
The additional analysis and discussions introduced during the rebuttal are valuable. I expect these to be properly integrated into the final manuscript.

**Key Questions For Authors:**

1.	Can you provide a formal mathematical justification for defining $\bigcup_{s_{t}\in\mathcal{S}}\mathcal{M}_{\theta}(s_{t})$ as a "manifold"? Given that the dynamics parameters are approximated by a neural network, what guarantees exist regarding the topological properties (e.g., continuity and differentiability) of this space? If it cannot be formally proven, please revise the terminology to "state-dependent subspaces" or similar.
2.	Could the authors provide more analysis or visualization of the learned action manifold to better understand what structure is being discovered?
3.	Could the authors provide real-world experiments to better evaluate the effectiveness of the proposed method?
4.	Could the authors comment on the potential of this method for humanoid systems, as well as the unique challenges that may arise in such settings?

**Limitations:**

1.	Given the limited experimental evaluation, it would be valuable for the authors to discuss the scope of the proposed method.
2.	Furthermore, the paper should discuss the limitations of MANIP when transitioning to real-world robots.

**Strengths And Weaknesses:**

- Strengths:
1.	The paper tackles the classical redundancy problem in motor control, which is an important problem in both robotics and neuroscience.
2.	The paper provides a closed-form solution for the optimal precoder based on a control-affine approximation of musculoskeletal dynamics.
3.	By relying on a dynamics model rather than extrinsic rewards, the method demonstrates robustness in sparse-reward environments. This is a valuable property for complex embodied tasks.
- Weaknesses:
1.	Manifold Definition: The mathematical rigor regarding the term "manifold" is lacking. The authors construct a state-dependent linear subspace $\mathcal{M}_{\theta}(s_{t})$ and state that the union of these subspaces across the state space $\bigcup_{s_{t}\in\mathcal{S}}\mathcal{M}_{\theta}(s_{t})$ forms a nonlinear manifold. However, without proving the continuity, differentiability, and local Euclidean properties of this union (which is highly dependent on the neural network parameterizing the dynamics), calling it a strict topological manifold is inaccurate.
2.	Limited experimental evaluation: The experiments focus on in-hand manipulation task suites, which makes it difficult to assess the generality of the proposed approach. Furthermore, the baselines are restricted to SAC, Lattice, and SAR. Need add more baselines to conclude whether the proposed method provides a general advantage.
3.	Writing: The introduction does not clearly articulate the specific problem the paper aims to address, nor does it sufficiently explain the key challenges faced by existing methods in this area. Further clarification would improve the readability and positioning of the work.

---

> ### Author Rebuttal · Authors · 2026-03-31
>
> We thank the reviewer for the rigorous critique of our terminology and for the constructive suggestions regarding the scope of our evaluation.
>
> > Manifold Terminology Revision
>
> We thank the reviewer for pointing out the topological nuances associated with the term manifold. Our intent was to highlight that the union of state-dependent subspaces enables a non-linear global control structure that exceeds the capacity of a single linear plane. We agree that formal topological properties like continuity and differentiability are not guaranteed by the neural network parameterization. We have revised the text as follows:
>
> "For any particular state, the subspace M_θ(s_t) is locally linear and low-dimensional. However, the union of these subspaces across the state space, ⋃_{s_t ∈ S} M_θ(s_t) ⊆ A, is not restricted to a single global linear subspace. This state-dependent parameterization allows the agent to access a non-linear action repertoire that can span the full action space A over time, providing significantly greater expressivity than a fixed, state-independent subspace."
>
> We have also updated the nomenclature of our framework to better reflect its information-theoretic core. We now refer to our objective as JoSE (Joint-Space Empowerment) and the resulting policy-learning algorithm as JoSEpi (formerly MANIP). We believe this refined branding, prompted by your feedback, provides a more memorable profile for the method while also avoiding the unintended topological connotations of manifold.
>
> > Action Manifold Analysis
>
> To investigate how control synergies evolve during learning and to align our findings with established motor neuroscience, we applied standard Electromyography (EMG) decomposition techniques to the muscle activations generated by our policy. Specifically, we used Non-negative Matrix Factorization (NMF) to extract muscle synergies at 8 equidistant intervals throughout training, evaluating the Variance Accounted For (VAF) by the components.
>
> Remarkably, we found that the VAF remains exceptionally stable throughout learning—shifting by only 5–10% between the 1st and 2nd iterations and remaining largely constant thereafter. This suggests that the fundamental "grammar" of the control space is established early, with subsequent iterations refining the usage of synergies rather than reinventing them.
>
> Furthermore, although JoSEpi operates within a 15D state-dependent subspace, traditional NMF analysis of the resulting muscle activations reveals that 15 synergies only account for 76–84% of the total variance. This discrepancy arises because our subspace is state-dependent; when marginalized across the full state space, the union of these local subspaces spans a higher-dimensional global volume. This finding carries significant implications for neuroscience: it suggests that the biological brain may utilize far fewer fundamental synergies than previous state-independent analyses have assumed, with the apparent complexity of motor behavior arising from the dynamic, state-contingent reorientation of low-dimensional control manifolds.
>
> We have added the NMF analysis and described its implications in the manuscript.
>
> >  Real-World Experiments
>
> By providing a mathematically grounded way to discover low-dimensional control subspaces (JoSE), we provide the software substrate required for physical hardware to function effectively. We hope that by demonstrating the success of JoSEpi on the highly complex MyoHand, we will inspire researchers who have access to physical robots to apply our subspace-discovery method to these systems. To facilitate this, we will publicly release our full implementation upon acceptance.
>
> We have added a dedicated Limitations section to the manuscript to discuss the specific challenges of sim-to-real transition (e.g., sensor noise and tendon friction).
>
> > Humanoid Systems
>
> Humanoid systems face unique challenges, notably the requirement to coordinate torques across the entire body (ankles, knees, hips, arms) to maintain balance. We believe JoSEpi is ideally-suited to this problem.
>
> Our framework can be applied to discover control subspaces that maximize an agent's control over its center-of-mass (CoM) rather than joint velocities. This approach would naturally identify synergies required to couple distal and proximal joints (e.g., ankle, knee and hip torques) to stabilize the CoM and achieve robust balance. Moreover, because the discovered control space is state-dependent, the synergies would naturally adapt to the mechanical requirements of the current state. For example, the model may prioritize fine-grained ankle strategies during quiet standing, but shift to hip and knee extensors to manage larger momentum shifts required during the forward step initiation.
>
> We have added this discussion to the manuscript, highlighting how our framework provides a principled, information-theoretic approach to isolating the relevant degrees of freedom for humanoid stability.

---

> > ### Author Rebuttal · Reviewer_TjYt · 2026-04-03
> >
> > Thank you for the detailed rebuttal. The revised terminology and renaming adequately address my concerns regarding the use of “manifold,” and I appreciate the effort to improve conceptual clarity.
> > The additional analysis and discussions introduced during the rebuttal are valuable. I expect these to be properly integrated into the final manuscript.

---

> > > ### Author Response · Authors · 2026-04-03
> > >
> > > Thank you for upgrading your score. We sincerely thank you for your thoughtful follow-up and for recognizing the improvements in clarity introduced by our revisions. We are glad that the updated terminology and additional analyses have addressed the concerns, and we appreciate the constructive guidance that helped strengthen the manuscript. We will ensure these contributions are fully and carefully integrated into the final version.

---

### Official Review · Reviewer_cmje · 2026-03-11

**Soundness:** 3
**Presentation:** 3
**Significance:** 3
**Originality:** 4
**Overall Recommendation:** 5
**Confidence:** 3

**Summary:**

This paper proposes Joint Space Empowerment (JSE) and its corresponding MANIP framework, formulating synergy discovery in musculoskeletal hands as an optimal precoding problem based on mutual information maximization, and demonstrates convincing empirical results on various dexterous manipulation tasks in MyoSuite.

**Compliance With Llm Reviewing Policy:**

Affirmed.

**Final Justification:**

I support this work because it grounds the discovery of synergies in high-dimensional musculoskeletal control within a clear, computable, and experimentally compelling information-theoretic framework, applying it to practical dexterous manipulation.

**Key Questions For Authors:**

1. The method in this paper is built on locally control-affine dynamics with approximately Gaussian noise, but in tasks like DieReorient and PenTwirl where contact is frequent and friction and contact switching are significant, the true dynamics often exhibit stronger nonlinearity, non-smoothness, or even multimodal characteristics. Could the authors further clarify: why is this approximation still sufficient to support the construction and effectiveness of JSE/MANIP in these contact-intensive tasks? Additionally, in the uncertainty characterized via MC dropout, is it possible that errors from model mismatch are mixed in, rather than purely epistemic uncertainty that aids exploration?




2. In the learning-to-learn experiment in Section 11.2, SAR appears to use a state-independent static linear subspace based on play/expert data, while MANIP uses a state-dependent dynamic manifold parameterization. If my understanding is correct, the performance difference here may conflate two factors: "the advantage of the JSE objective" and "the stronger representational capacity of state-dependent representation." Could the authors further clarify whether this comparison is fair in terms of representational capacity, and consider adding an ablation that maintains the state-dependent manifold structure but does not use JSE, to help more clearly attribute the source of performance improvement?

**Limitations:**

Yes

**Strengths And Weaknesses:**

### Strengths




1.  The paper does not stop at empowerment over the general state space, but further narrows down to joint-space/functional DOF, framing synergy or action manifold discovery as maximizing mutual information between latent actions and controlled joint variables. This framing combines biological inspiration, control, and representation learning, making it an attractive entry point overall.




2. The paper derives a closed-form optimal precoder under affine-Gaussian assumptions and connects the optimal subspace to the right singular vectors of the whitened control matrix. Even if there is room for further discussion on practical implementation, this connection from information-theoretic objectives to computable structure is valuable in itself.




3. MyoHand/MyoSuite is a representative musculoskeletal hand manipulation environment, and tasks such as BaodingBalls, DieReorient, KeyTurn, and PenTwirl cover challenging dexterous manipulation scenarios, lending weight to the experimental results at the application level.




### Weaknesses




1. A gap between theoretical conclusions and practical implementation remains insufficiently clarified. The theorem provides a closed-form optimal solution for a specific precoder, but the actual implementation uses a different, more practical basis. According to Appendix A.5, my understanding is that the authors mainly prove that this practical basis spans the same subspace as the theoretical solution, but do not further explain whether it still retains the same capacity optimality as Theorem A.1 under the actual parameterization. Therefore, the current theoretical result is more of a characterization of the optimal subspace, while the actual algorithm uses a practical surrogate; the statement about "closed-form optimal solution" in the main text could perhaps be more precise.




2. The optimality argument in Theorem A.1 could be stated more completely. The current proof clearly explains why $W_r$ is a natural construction and why the optimal covariance should be diagonal in that coordinate system; however, from the exposition, it is not sufficiently clear whether this construction covers all cases of the global optimal solution, and the non-uniqueness of solutions (e.g., equivalent rotation classes) is not adequately addressed. Although this paper is not a theory paper and this does not constitute a fundamental issue, if the authors could further supplement the scope of optimality, equivalent solution classes, and their relationship to the actual parameterization, the theoretical part would be more rigorous.

---

> ### Author Rebuttal · Authors · 2026-03-31
>
> We thank reviewer cmje for their insightful summary.
>
> > Control-Affine Accuracy
>
> We hypothesize that the control-affine structure can in principle accommodate state-dependent discontinuities without large model errors, as the transition dynamics can be arbitrarily nonlinear with respect to the state.
>
> To assess how well the control-affine approximation holds during manipulation, we examined one-step prediction errors of the learned dynamics model during contact events. We group mean squared prediction errors based on whether at least one finger tip transitioned from contact to no contact or vice versa (Contact Transition) across consecutive timesteps.
>
> The pooled dataset (N = 273,940 observations) showed that the mean squared error (MSE) for a Contact Transition was 0.172 (SEM = 0.001), compared to 0.150 (SEM = 0.001) for states with no Contact Transition (Table 1). The standard deviations in both conditions (0.334 and 0.466, respectively) are large relative to the difference in means, indicating substantial variability within each group.
>
> #### Table 1: MSE
>
> | Contact Transition   |   Mean |   SEM |   Count |   STD |
> |:---------------------|-------:|------:|--------:|------:|
> | No                   |  0.15  | 0.001 |   93825 | 0.334 |
> | Yes                  |  0.172 | 0.001 |  180115 | 0.466 |
>
>
> To formally evaluate this difference, we employed a linear mixed-effects model with a fixed effect of contact transition and random intercepts for each experimental run (environment–seed combination).
>
> The mixed-effects model revealed a statistically significant effect of contact transitions on mean squared error (β = 0.056, SE = 0.002, z = 31.38, p < 1 × 10⁻¹⁶) (Table 2). This result should be interpreted in the context of the very large sample size (N = 273,940 observations), where even small differences can achieve strong statistical significance.
>
> #### Table 2: Mixed Effects Model Results
> |                    |   Coef. |   Std. Err. |      z | P           |   95% CI (2.5%) |   95% CI (97.5%) |
> |:-------------------|--------:|-----------:|-------:|:------------|----------------:|-----------------:|
> | Intercept          |   0.142 |      0.036 |  3.996 | 6.5 x 10⁻⁰⁵ |           0.073 |            0.212 |
> | Contact Transition |   0.056 |      0.002 | 31.382 | < 1 x 10⁻¹⁶ |           0.052 |            0.059 |
>
>
> The modest increase in prediction error suggests that performance remains largely stable across task phases, with limited degradation during contact transition events. Given that the transition dynamics can be arbitrarily nonlinear with respect to the state observation, we speculate that with richer observations (e.g. haptic or force feedback), the prediction errors during contact could be reduced further.
>
> > MC Dropout Uncertainty and Model Mismatch
>
> We do not think that MC dropout uncertainty will have errors from model mismatch mixed in, as errors from model mismatch are irreducible, and MC dropout uncertainty represents reducible uncertainty. With model mismatch, different networks sampled from the dropout ensemble will converge to the same biased one-step prediction, reaching full agreement and exhibiting no epistemic uncertainty.
>
> > Fairness of Comparison in Learning-to-Learn
>
> To isolate the benefit of the JSE objective versus state-dependent representational capacity, we compare MANIP against a State-Dependent SAR (SD-SAR) baseline.
>
> We train a state-conditioned linear autoencoder on play-phase data.
>
> An orthonormal decoder matrix  $D(s; \theta) \in \mathbb{R}^{n \times k}$ is derived by passing the state $s$ through a neural network with weights $\theta$ to produce a raw matrix $V(s; \theta)$. We then apply Modified Gram-Schmidt (MGS) to the columns of $V(s; \theta)$ to construct $D(s; \theta)$.
>
> Rather than train an encoder network, we compute the optimal linear encoder in closed form. For an orthonormal decoder $D(s; \theta)$, the latent variable $z_{opt}$ that minimizes the Euclidean reconstruction loss $\|x - D(s; \theta)z\|^2$ is given by the projection:$$z_{opt} = D(s; \theta)^\top x$$
>
> The baseline is trained by optimizing $\theta$ to minimize reconstruction error :$$\mathcal{L}(\theta) = \| x - D(s; \theta) D(s; \theta)^\top x \|^2$$
>
> MANIP significantly outperforms SD-SAR (Table 3), confirming that the performance gains in MANIP are not merely the result of increased representational capacity.
>
> #### Table 3: Mean ± SEM Fraction Solved
>
> | Model | Reorient100 | ReorientID | ReorientOOD |
> | :--- | :---: | :---: | :---: |
> | **MANIP** | 0.511 ± 0.020 | 0.477 ± 0.019 | 0.336 ± 0.010 |
> | **SAR** | 0.407 ± 0.021 | 0.361 ± 0.019 | 0.261 ± 0.016 |
> | **SD-SAR** | 0.353 ± 0.044 | 0.337 ± 0.031 | 0.236 ± 0.024 |
> | **SAC** | 0.307 ± 0.062 | 0.254 ± 0.053 | 0.178 ± 0.036 |
>
> By capturing patterns of muscle coordination from the play-phase task, reconstruction-based methods are inherently task-specific. In contrast, the dynamics-based JSE objective is task-agnostic and so generalizes better across tasks.

---

> > ### Author Rebuttal · Reviewer_cmje · 2026-04-02
> >
> > Thank you for the rebuttal. I will keep my positive rating.

---

> > > ### Author Response · Authors · 2026-04-02
> > >
> > > Thank you for maintaining your support of our paper. We are very grateful for your insightful critiques. We will ensure that the results from the new experiment proposed by you are included in the manuscript, and we will incorporate our answers to your key questions. The final paper will be significantly stronger thanks to your contribution.

---

### Decision · Program_Chairs · 2026-04-30

**Decision:**

Accept (spotlight)

**Comment:**

This paper proposes Joint-Space Empowerment (JSE) and the MANIP algorithm, an information theoretic framework for identifying a low-dimensional action manifold for improving learning effectiveness and efficiency for over-actuated musculoskeletal systems.  Reviewers generally agreed the method solved an important problem using an elegant and effective point of view.  They highlighted the strong theoretical derivation of a closed form solution for optimal precorder in this setting.  The method is backed up by strong experimental results on challenging manipulation tasks.  Reviewers noted some clarity issues, in particular in the relationship between the derivation and practical implementation, and ways the experimental evaluation could be expanded.  The authors did a good job addressing these concerns and are encouraged to revise their submission accordingly.